# MPP8 is essential for sustaining self-renewal of ground-state pluripotent stem cells

Iris Müller [1,2,3], Ann Sophie Moroni [1,2], Daria Shlyueva[1,2,3], Sudeep Sahadevan[1,2], Erwin M. Schoof [4,5], Aliaksandra Radzisheuskaya[1,2,3], Jonas W. Højfeldt [1,2], Tülin Tatar[1,2], Richard P. Koche [6], Chang Huang[3] & Kristian Helin [1,2,3 ✉]

Deciphering the mechanisms that control the pluripotent ground state is key for understanding embryonic development. Nonetheless, the epigenetic regulation of ground-state mouse embryonic stem cells (mESCs) is not fully understood. Here, we identify the epigenetic protein MPP8 as being essential for ground-state pluripotency. Its depletion leads to cell cycle arrest and spontaneous differentiation. MPP8 has been suggested to repress LINE1 elements by recruiting the human silencing hub (HUSH) complex to H3K9me3-rich regions. Unexpectedly, we find that LINE1 elements are efficiently repressed by MPP8 lacking the chromodomain, while the unannotated C-terminus is essential for its function. Moreover, we show that SETDB1 recruits MPP8 to its genomic target loci, whereas transcriptional repression of LINE1 elements is maintained without retaining H3K9me3 levels. Taken together, our findings demonstrate that MPP8 protects the DNA-hypomethylated pluripotent ground state through its association with the HUSH core complex, however, independently of detectable chromatin binding and maintenance of H3K9me3.

[1] Biotech Research and Innovation Centre (BRIC), Faculty of Health and Medical Sciences, University of Copenhagen, Copenhagen, Denmark. [2] The Novo Nordisk Foundation for Stem Cell Biology (DanStem), Faculty of Health and Medical Sciences, University of Copenhagen, Copenhagen, Denmark. [3] Cell Biology Program and Center for Epigenetics Research, Memorial Sloan Kettering Cancer Center, New York, NY, USA. [4] The Finsen Laboratory, Rigshospitalet and Biotech Research and Innovation Centre (BRIC), Faculty of Health Sciences, University of Copenhagen, Copenhagen, Denmark. [5] Department of Biotechnology and Biomedicine, Technical University of Denmark, Lyngby, Denmark. [6] Center for Epigenetics Research, Memorial Sloan Kettering Cancer Center, New York, NY, USA. ✉email: helink@mskcc.org

Embryonic stem cells (ESCs) are pluripotent cells derived from the inner cell mass of the preimplantation embryo. They can indefinitely self-renew in culture and are capable of differentiating into all somatic lineages[1]. Traditionally, mouse ESCs (mESCs) were maintained in medium containing serum and the cytokine leukemia inhibitory factor (LIF)[2,3]. More recently, combinatorial inhibition of MEK/ERK and GSK3 signaling pathways using two small-molecule inhibitors (2i), PD0325901 and CH99021, respectively, yielded homogenously high expression levels of pluripotency transcription factors[4]. Hence, these culture conditions more closely reflect so-called ground-state pluripotency. While the role of transcription factors in maintaining self-renewal is well established[5,6], the epigenetic regulation of the pluripotent ground-state remains incompletely understood. However, genome-wide studies of epigenetic modifications of chromatin[7] and proteomic profiling of chromatin-associated complexes and histone modifications[8] have revealed distinct features of the ground-state epigenome, indicating a unique contribution of epigenetics for ground-state pluripotency.

Here, we report on the use of CRISPR/Cas9 screening to identify mediators of ground-state pluripotency of mESCs grown in 2i/LIF. We show that the chromodomain protein M-Phase Phosphoprotein 8 (MPP8) is selectively required for ground-state pluripotent stem cells survival and demonstrate that MPP8 is essential for safeguarding ground-state pluripotency. MPP8 has two annotated domains: an N-terminal chromodomain, which has been shown to bind H3K9me3 in vitro and in vivo[9–11] and four consecutive ankyrin-repeat domains towards its C-terminus with unknown function. Moreover, MPP8 has been shown to interact with multiple epigenetic silencing proteins, including the H3K9 mono- and di-methyltransferase proteins GLP/G9a[11,12], DNA methyltransferase DNMT3A[11,12], histone deacetylase SIRT1[13] and ATF7IP, a known binding partner of H3K9 tri-methyltransferase SETDB1[14]. Recently, MPP8 was found to form the human silencing hub (HUSH) core complex together with TASOR and PPHLN1 and SETDB1 as an associated catalytic subunit[15]. It was suggested that the core complex recruits SETDB1, enabling the propagation of H3K9me3 from a heterochromatin-adjacent region onto a transgene leading to its silencing, an effect known as position-effect variegation[15]. Other studies have suggested that the HUSH complex represses endogenous LINE1 elements[16] through a mechanism involving the recruitment of TASOR to ERV elements via KAP1 (TRIM28) and the subsequent spreading of H3K9me3 to adjacent LINE1 elements via cycles of reading and writing of H3K9me3 by MPP8 and SETDB1, respectively[17].

In this study, we show that the previously uncharacterized C-terminal part of MPP8 is key for mESC self-renewal and that is required for the binding to the HUSH-complex member TASOR. Cells expressing only the chromodomain-containing N-terminal part of MPP8 show HUSH core complex destabilization, loss of MPP8 chromatin binding, and increased expression of LINE1 elements. Unexpectedly, the N-terminal part of MPP8 is dispensable for the maintenance of stem cell identity. MPP8 mutants lacking the chromodomain display efficient repression of LINE1 elements, despite losing detectable binding to chromatin and SETDB1. We further demonstrate that SETDB1 recruits MPP8 to its genomic target sites to initiate LINE1 repression. However, once repression is established, H3K9me3 is not required. In summary, these results suggest an alternative mechanism by which MPP8 participates in the repression of LINE1 elements, which requires its association with the HUSH core complex but is independent of its binding to H3K9me3 and SETDB1.

## Results

**Identification of essential epigenetic regulators in mESCs.** To identify proteins essential for mESC self-renewal, we performed a CRISPR/Cas9 screen using a curated sgRNA library targeting functional domains of 1218 proteins involved in epigenetic processes encoded in the mouse genome. The targeting of functional domains has previously been demonstrated to yield enhanced gene inactivation in negative selection screens[18]. To construct the library, we manually classified PFAM-annotated domains into five categories and preferentially selected sgRNAs targeting either catalytic, epigenetic reader, or DNA-binding domains. The final library was composed of a total of 12,472 sgRNAs, including 981 non-targeting sgRNAs from the mouse GeCKO library[19] to constitute roughly 10% of the total library and sgRNAs targeting 20 positive control genes, such as the pluripotency gene *Nanog* and the housekeeping gene *Pcna*. For the majority of genes 10 or 11 sgRNAs were generated, most of them targeting catalytic domains (35.3%) followed by DNA-binding domains (27.1%) and epigenetic reader domains (14.7%) (Supplementary Fig. 1a, b).

We engineered mESCs expressing Cas9 and validated its activity by transduction of lentiviruses expressing sgRNAs targeting different regions of *Pou5f1* (encoding OCT4) or non-targeting controls (Supplementary Fig. 1c–f). For the negative selection screen, cells were harvested after 10 and 15 population doublings (days seven and eleven, respectively), and the abundance of individual sgRNAs was subsequently assessed by Next Generation Sequencing (Fig. 1a). Comparison of sgRNA abundance between day one and day seven or day eleven, respectively, revealed that non-targeting control sgRNAs were not depleted, while the majority of sgRNAs targeting positive control genes, such as *Pou5f1* or *Polr1c*, were depleted more than 16-fold. These results show that the dropout screening was both specific and efficient (Fig. 1b).

We defined hits as genes that were at least 10-fold depleted by at least one sgRNA comparing day 11 to day 0 of the screen. Using these criteria, we identified 146 genes encoding proteins with potential epigenetic function as contributing to the self-renewal of ground-state mESCs. Among these, several genes had previously been shown to be essential for mESC self-renewal (Supplementary Table 1), including members of the Tip60-p400 complex[20]. This finding further validates the relevance of the screen. The majority of the 146 hits overlapped with genes previously identified in genome-wide screens in mESCs grown in serum/LIF[21] (109 genes) or 2i/LIF[22] (105 genes) culture conditions (Fig. 1c). As our screen did not discriminate between loss of stem cell self-renewal or pathways implicated in general house-keeping processes, we compared our hits with genes classified as common essential through genome-wide dropout screens using 739 cancer cell lines[23]. This analysis suggested that 117 of the 146 genes are essential for general cell proliferation and survival (Fig. 1c).

Eight genes specifically caught our attention as these had neither been identified in previous mESCs CRISPR/Cas9 screens nor flagged as common essential (Fig. 1c). However, among these there were genes that had been shown to have crucial roles in pluripotency, such as *Ash2l*[24,25] and *Lin9*[26,27]. Next, we selected two genes which had not previously been associated with stem cell self-renewal, namely *Mphosph8* (encoding MPP8) and *Hlcs*. We reasoned that potential factors selectively required for self-renewal of ground-state pluripotent cells would be essential for ground-state mESCs while not being required for cells derived from later embryonic stages. To test the specific requirement of the two selected genes for mESCs, we then compared their impact on cell growth in mESCs and mouse embryonic fibroblasts (MEFs) in competition-based proliferation

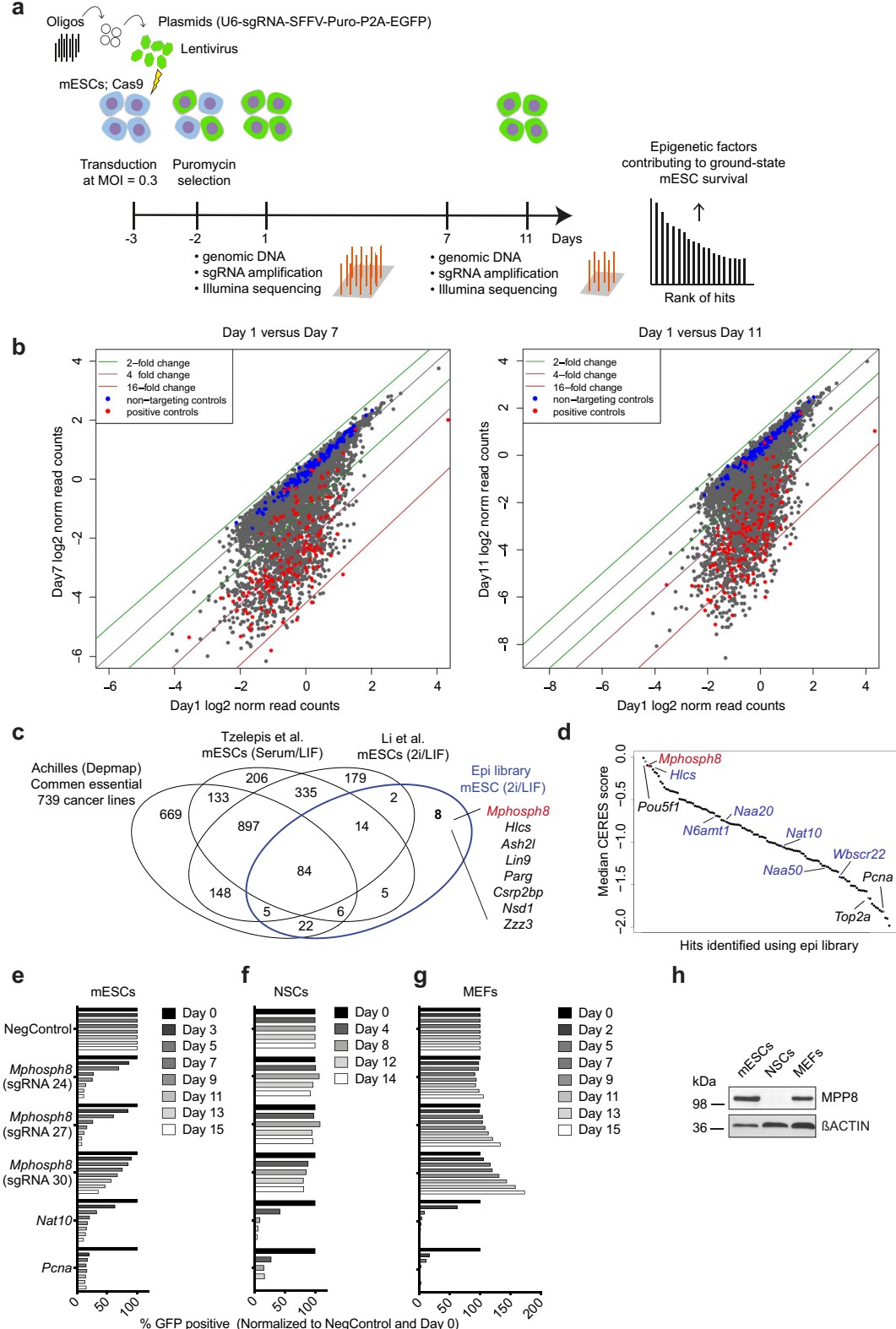

assays (Supplementary Fig. 2a, b). To test the reliability of cancer cell-derived CERES scores for general essentiality in physiological processes we also included five genes identified in our screen, *N6amt1*, *Wbscr22*, *Nat10*, *Naa20* and *Naa50*, which had not previously been studied in ground-state pluripotent mESCs but showed noteworthy depletion across the 739 cancer cell lines tested (Fig. 1d). The individual targeting of all the seven selected

genes impaired the growth of mESCs, validating the screening results (Supplementary Fig. 2a). Consistent with being required for the proliferation of many cell lines, the deletion of *N6amt1*, *Naa20*, *Nat10*, *Naa50*, and *Wbscr22* also affected proliferation of MEFs, although to different extents (Supplementary Fig. 2b). Depletion of *Hlcs*, which has not been classified as a common essential, mildly affected MEF proliferation. *Mphosph8* was the

**Fig. 1 Domain-focused CRISPR/Cas9 screen identifies potential regulators of epigenetic processes contributing to self-renewal of ground-state mESCs in 2i/LIF. a** Schematic representation of the CRISPR/Cas9 screening strategy. All sgRNAs were cloned into the pU6-sgRNA-SFFV-Puro-P2A-EGFP vector. mESCs were transduced at a multiplicity of infection (MOI) of ~0.3. Cells were passaged every second day, pellets were collected at day 1, day 7 (10 population doublings) and day 11 (15 population doublings) of the screen and the abundance of the individual sgRNAs was determined. **b** Scatterplot comparing day 7 and day 11, respectively, to day 1 log2 median-normalized read counts of the individual sgRNAs. **c** Overlap analysis of mouse homologues of common essential genes in 739 human cancer cell lines (Project Achilles, Depmap, 20Q1[23]), 1680 and 1664 genes identified as essential for mESCs grown in serum/LIF[21] or 2i/LIF[22], respectively using genome-wide sgRNA screens, and the 146 genes encoding proteins with potential epigenetic function identified in this study. **d** Ranked comparison of median CERES scores (Project Achilles, Depmap, 20Q1[23]) for human genes and for the genes identified in this study. CERES scores provide estimates of gene dependency with values lower than −0.5 representing noteworthy depletion across the 739 cancer cell lines tested and hence the gene is classified as common essential[23]. **e–g** Competition-based proliferation assays in indicated Cas9-expressing cell lines. GFP was monitored over a time course of more than ten population doublings. sgRNAs targeting the core essential genes *Nat10* and *Pcna* served as positive controls while a non-targeting sgRNA served as a negative control (NegControl). The percentage of GFP+ cells is normalized to the day 0 measurement and the measurement of the NegControl at the respective day ($n = 1$). **h** Western blot analysis of MPP8 and ßACTIN (loading control) in the indicated cell lines ($n = 1$). mESCs = mouse embryonic stem cells; NSCs = neural stem cells; MEFs = mouse embryonic fibroblasts. Source data are provided as a Source Data file.

only candidate gene that selectively impaired propagation of mESCs: removal in MEFs caused a positive selection phenotype, whereas no growth defect was observed in neural stem cells (NSCs, Fig. 1e–g). Western blot analysis confirmed that MPP8 is expressed in mESCs and MEFs and absent in NSCs (Fig. 1h).

Hence, we concluded that ground-state pluripotent mESCs are selectively dependent on MPP8, and we therefore chose to focus on elucidating the specific role of MPP8 in regulating self-renewal of these cells.

**MPP8 is essential for self-renewal of ground-state mESCs.** To determine the impact of MPP8 loss on the maintenance of mESC identity, we established a drug-inducible MPP8 depletion system by tagging its C-terminus with a miniAID degron and over-expressing OsTIR1 in mESCs (Fig. 2a). In this system, exogenously expressed OsTIR1 forms a functional ubiquitin ligase complex with cell endogenous components (SKP1 and CUL1), which dimerizes with miniAID-tagged proteins upon addition of auxin, allowing for their rapid degradation[28,29]. In absence of auxin, the miniAID tag itself did not lead to lower expression of MPP8 or to inhibition of MPP8 function, since the cells were completely viable and proliferated normally (Supplementary Fig. 3a, b). Assessment of the degradation kinetics revealed a complete depletion of MPP8$^{mAID}$ in OsTIR1-expressing cells already 1 h after auxin treatment (Fig. 2b). Interestingly, while removal of MPP8 only led to a slightly reduced proliferation in serum/LIF culture conditions, cell death was induced rapidly in 2i/LIF culture conditions (Fig. 2c, d).

We next investigated whether depletion of MPP8 affected cell cycle progression. Strikingly, we observed a 3-fold increased percentage of cells in G1 phase, concomitant with a 2-fold reduced percentage of cells in the S phase at 72 h post-auxin treatment (Fig. 2e). Next, we asked whether stem cell self-renewal was compromised through increased differentiation upon MPP8 depletion in the metastable stem cell state in serum/LIF. Alkaline phosphatase staining showed a significant increase of differentiated colonies by more than 2-fold. (Fig. 2f), consistent with rapidly induced cell death in 2i/LIF as differentiated cells cannot be maintained under these culture conditions. We further noticed a drastic reduction of colony numbers under differentiation permissive culture conditions upon MPP8 depletion, suggesting an involvement of MPP8 in mESC differentiation (Fig. 2g). However, when monitoring the early phase of mESC transition after withdrawal from 2i in serum-free N2B27 medium (Fig. 2h), MPP8 loss led to a fast destabilization of the naïve gene expression program (Fig. 2i), while no accelerated upregulation of differentiation markers was observed (Fig. 2j). These results suggest that the primary effect of MPP8 is to promote self-

renewal of naïve mESCs. Furthermore, MPP8-depleted mESCs cultured in serum/LIF in the presence of GSK inhibitor alone were viable, but not when cultured in the presence of MEK inhibitor alone or both inhibitors (Supplementary Fig. 3c, d). These results indicate that the activity of the MEK/ERK signaling pathway overomes the requirement of MPP8 in mESCs.

In summary, our results demonstrate that MPP8 is essential for stem cell self-renewal of ground-state pluripotent mESCs by maintaining the proliferative state.

**MPP8's C-terminal region is mediating its essential function.** To understand the mechanism by which MPP8 contributes to self-renewal of ground-state pluripotent mESCs, we performed complementation analysis in MPP8-depleted mESCs. To do this, we ectopically expressed either full-length or mutated *Mphosph8* cDNAs resistant to Cas9 targeting by sgRNA 24 in mESCs (Fig. 3a, b). All constructs were successfully expressed as assessed by qPCR (Supplementary Fig. 4a). For the chromodomain mutant we mutated all three residues forming the aromatic cage into alanine (MPP8$^{F59A;W80A;Y83A}$)[12,30], while for the ankyrin-repeat mutant we removed the complete domain (MPP8$^{ΔARD}$) as no structural information was available. The ability of the mutant MPP8 proteins to rescue the lack of endogenous MPP8 was subsequently tested in competition-based proliferation assays (Fig. 3c). To our surprise, both mutants rescued the observed growth defect to the same extent as wild-type MPP8, demonstrating that none of the two defined domains of MPP8 are required for its ability to maintain mESC self-renewal (Fig. 3c).

To further delineate the function of N- and C-terminal parts, we engineered additional mutants with deletions in either MPP8's N-terminal part (MPP8$^{112–858}$) or C-terminal part (MPP8$^{1–729}$, MPP8$^{1–522}$, MPP8$^{1–188}$). While the N-terminal deletion mutant could rescue the phenotype, all three C-terminal deletion mutants failed to do so (Fig. 3c). To corroborate these results by an orthogonal approach we overexpressed the same MPP8 mutants in the *MPP8-mAID* targeted mESCs. Ectopic protein expression levels were comparable to endogenous expression levels as well as similar between all ectopically expressed proteins, apart from MPP8$^{1–522}$ and MPP8$^{1–188}$, which displayed enhanced protein stability (Supplementary Fig. 4b). Addition of auxin led to efficient degradation of endogenous MPP8 while exogenous expression remained stable (Supplementary Fig. 4c). Again, MPP8$^{wt}$, MPP8$^{F59A;W80A;Y83A}$, MPP8$^{ΔARD}$, and MPP8$^{112–858}$ rescued growth defects caused by MPP8 degradation while all three C-terminal mutants failed to do so (Fig. 3d). Based on these results, we conclude that the very C-terminal part of MPP8, but not the N-terminal part including the chromodomain, is crucial for maintaining the self-renewal of ground-state mESCs.

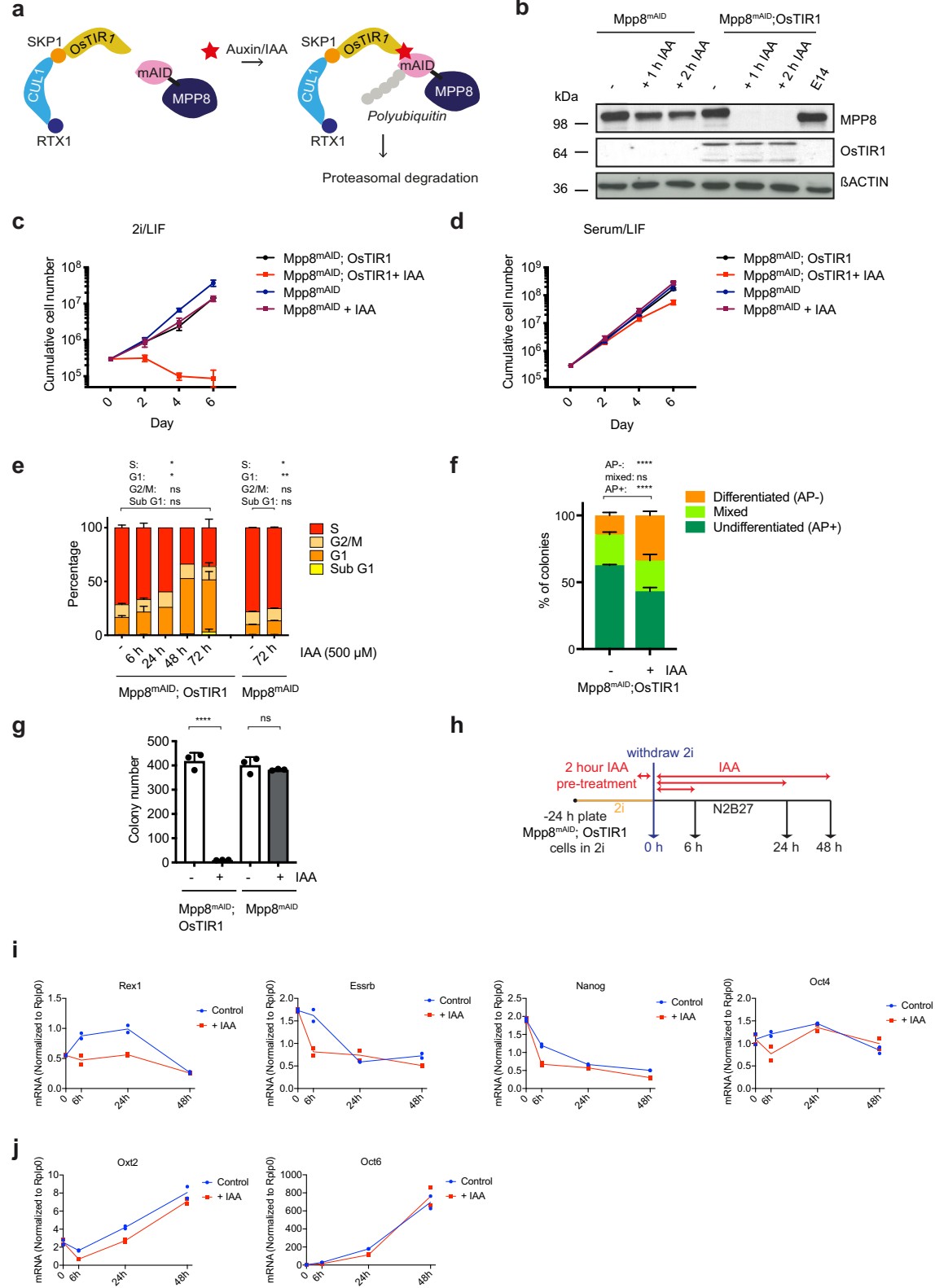

**MPP8 requires stable HUSH core complex interaction in mESCs.** The requirement of MPP8's uncharacterized C-terminal part for its essential function prompted us to ask which proteins are bound to this region. Such interactions could potentially explain a chromodomain-independent self-renewal mechanism. To address this, we affinity-purified proteins associated with FLAG-tagged wild-type MPP8 and N-terminal and C-terminal

mutants of MPP8 and determined the identity of the associated proteins by liquid chromatography tandem mass spectrometry (LC–MS/MS) (Fig. 4a). For wild-type MPP8 we observed high enrichment of both the HUSH members TASOR and SETDB1 in the presence of 300 mM NaCl, indicating that these are stable interactors of MPP8 in mESCs (Fig. 4b). In these experiments we also detected low, but significant enrichments, of the third HUSH

**Fig. 2 MPP8 is essential for stem cell self-renewal of ground-state mESCs. a** Schematic representation of the auxin-inducible degradation system for the MPP8 protein. **b** Kinetic evaluation of Mpp8[mAID] degradation in OsTIR1-expressing mESCs grown in serum/LIF after 1 or 2 h of auxin treatment (+ IAA) in Mpp8[mAID]; OsTIR1 or Mpp8[mAID] (control) cells ($n = 1$). **c, d** Cell proliferation assay in 2i/LIF (**c**) or serum/LIF (**d**) culture conditions ± IAA (500 μM) (mean ± s.d., $n = 3$ biological replicates). **e** Cell cycle analysis in 2i/LIF culture conditions ± IAA (500 μM) (mean ± s.d., $n = 3$ independent experiments for untreated, 6 h, 72 h; mean, $n = 2$ independent experiments for 24 h, 48 h). $p = 0.0136$ (S), $p = 0.0169$ (G1), $p = 0.7009$ (G2M), $p = 0.1793$ (SubG1) comparing each IAA-treated (72 h) cell population to the respective untreated cell population of Mpp8[mAID]; OsTIR1 cells and $p = 0.0209$ (S), $p = 0.0014$ (G1), $p = 0.2478$ (G2M), $p = 0.6382$ (SubG1) comparing each IAA-treated (72 h) cell population to the respective untreated cell population of Mpp8[mAID] cells (two-tailed paired Student's $t$ test). **f** Quantification of the Alkaline Phosphatase (AP) staining assay in serum/LIF culture conditions. Stained colonies were scored based on AP+ (undifferentiated), mixed or AP- (differentiated) morphology (mean ± s.d., $n = 3$ independent experiments). $p = 2.0036e{-}06$ (AP+), $p = 0.6763$ (mixed) and $p = 1.8854e{-}06$ (AP-) comparing each IAA-treated subgroup to the respective untreated subgroup. (We report the maximum $p$ value of pairwise two-sample t-tests (two-tailed), that is each replicate of untreated against each replicate of auxin-treated cells). **g** Quantification of colony formation. Differentiation was induced by removal of LIF under serum-containing culture conditions (mean ± s.d., $n = 3$ independent experiments). $p = 2.8985e{-}05$ for Mpp8[mAID]; OsTIR1 and $p = 0.3808$ for Mpp8[mAID] (two-tailed unpaired Student's $t$ test). **h** Experimental setup for monolayer differentiation of naïve mESCs in N2B27 by withdrawal of 2i. **i, j** qRT-PCR analysis of selected pluripotency genes (**i**) or early post-implantation epiblast markers (**j**) ($n = 2$ biologically independent samples). *$p < 0.05$, **$p < 0.01$, ***$p < 0.001$, ****$p < 0.0001$, ns = not significant. Source data are provided as a Source Data file.

core complex component, PPHLN1, as well as associated member ATF7IP, which is a well-studied direct interactor of SETDB1[31]. The lower enrichment of PPHLN1 and ATF7IP with MPP8 could potentially indicate that the proteins indirectly bind via TASOR and SETDB1, respectively. In agreement with this, both PPHLN1 and MPP8 were detected, when TASOR-associated proteins were purified (Supplementary Fig. 5a), suggesting that PPHLN1 interacts with MPP8 through the binding to TASOR.

For the C-terminally truncated MPP8 the interaction with SETDB1 was maintained, while binding to TASOR was lost. This result shows that the interaction between MPP8 and TASOR is dependent on the C-terminal part of MPP8 and indicates that the integrity of the HUSH core complex is essential for mESC survival (Fig. 4c). In fact, we observed a decrease in TASOR protein levels upon degradation of MPP8, which is in agreement with previous observations in HeLa cells[15]. Moreover, we showed that the stability of TASOR is dependent on the C-terminal part of MPP8, which is required for the binding to TASOR (Supplementary Fig. 5b). Interestingly, the affinity purification results also showed that the binding of MPP8 to SETDB1 was strongly impaired for mutants of MPP8 lacking the chromodomain, suggesting that this interaction is not required for MPP8 to maintain mESCs in the pluripotent ground state (Fig. 4d). In summary, our results show that the N-terminus of MPP8 is required for the binding to SETDB1 and the C-terminus for the binding to TASOR and PPHLN1. Hence, the interaction with TASOR and PPHLN1 is essential for the proliferation of mESCs, whereas the binding to SETDB1 is not.

To further investigate the involvement of other HUSH complex proteins for the proliferation of ground-state mESCs, we performed competition-based proliferation assays. Targeting of both core complex members, *Pphln1* and *Tasor*, as well as HUSH-associated members, *Setdb1*, *Atf7ip,* and *Morc2a*, affected proliferation in 2i/LIF conditions (Fig. 4e). In contrast, the HUSH core complex members were only mildly impacting cell proliferation of mESCs grown in serum/LIF conditions (Fig. 4e). Moreover, MEFs did not show any proliferative disadvantage compared to wild-type cells, when expressing sgRNAs to *Pphln1* and *Tasor* (Fig. 4e). GLP, which was not identified in our interactome analysis but has been proposed to interact with MPP8[11,12], was essential in both cell lines. These results further indicate that the essential function of MPP8 in ground-state mESCs is mediated through its interaction with proteins in the HUSH complex.

To obtain mechanistic insights into why the HUSH complex is essential for the self-renewal of ground-state mESCs, we determined its chromatin binding pattern. To do this, we first demonstrated that we could identify DNA regions specifically enriched for MPP8 in mESCs (Supplementary Fig. 5c). Then, we determined the genomic localization of MPP8 in mESCs using ChIP-seq, which led to the identification of 55 regions significantly enriched by wild-type MPP8 that were not identified in MPP8 knockout mESCs (Supplementary Fig. 5d). In agreement with previously published results[15,16], MPP8 binding sites were co-enriched for H3K9me3 (Supplementary Fig. 5d). The majority of MPP8 was found to be located in intergenic regions (60%) followed by introns (22%), and transcription termination sites (11%) (Supplementary Fig. 5e).

To determine if the other members of the HUSH complex are enriched at MPP8-bound regions, we created mESC lines in which endogenous PPHLN1 and TASOR were C-terminally tagged with a double FLAG epitope (Supplementary Fig. 5f). In agreement with being partners of MPP8 in the HUSH complex, both proteins were found to associate with MPP8-bound regions (Fig. 5f and Supplementary Fig. 5g). Moreover, MPP8-bound regions significantly overlapped with previously published MORC2A binding sites in mESCs[32], as previously shown in HeLa cells[33], further supporting a potential role of MORC2A in HUSH-mediated silencing at these loci in mESCs (Fig. 4f and Supplementary Fig. 5h, i).

Subsequently, we examined the genomic localization of MPP8 mutants by ChIP-seq. Strikingly, both the N-terminal and the C-terminal part of MPP8 were required for detectable binding of MPP8 to chromatin (Fig. 4g, h). ChIP-qPCR validation further corroborated these results (Supplementary Fig. 6a). Although the polyclonal MPP8 antibody employed in the ChIP-assays can recognize both mutants in western blot (Supplementary Fig. 4b), we used FLAG immunoprecipitation in the ChIP for ectopic MPP8[wt], MPP8[1–729] and MPP8[112–858] to rule out potential differences in affinity of the antibody to the different proteins. This complementary approach confirmed the binding of wild-type MPP8 to high-confidence target sites and that both MPP8 mutants had lost this ability (Fig. 4g, i). These results were further validated by ChIP-qPCR (Supplementary Fig. 6b).

While the loss of enrichment for MPP8[112–858] could reflect the failure of MPP8 to bind chromatin without its chromodomain, the lack of detectable chromatin binding of MPP8[1–729] shows that the chromodomain is not sufficient for the stable binding of MPP8 to chromatin. Moreover, it shows that the C-terminal part of MPP8 is required for the stable binding of MPP8 to chromatin, which could potentially be mediated by the other members of the HUSH complex. To test if the HUSH complex could still bind to its canonical target sites in absence of the chromodomain of MPP8, we inserted a sequence coding for a Flag tag into the *Tasor* locus of mESCs harboring mini-AID tagged MPP8 and OsTIR1.

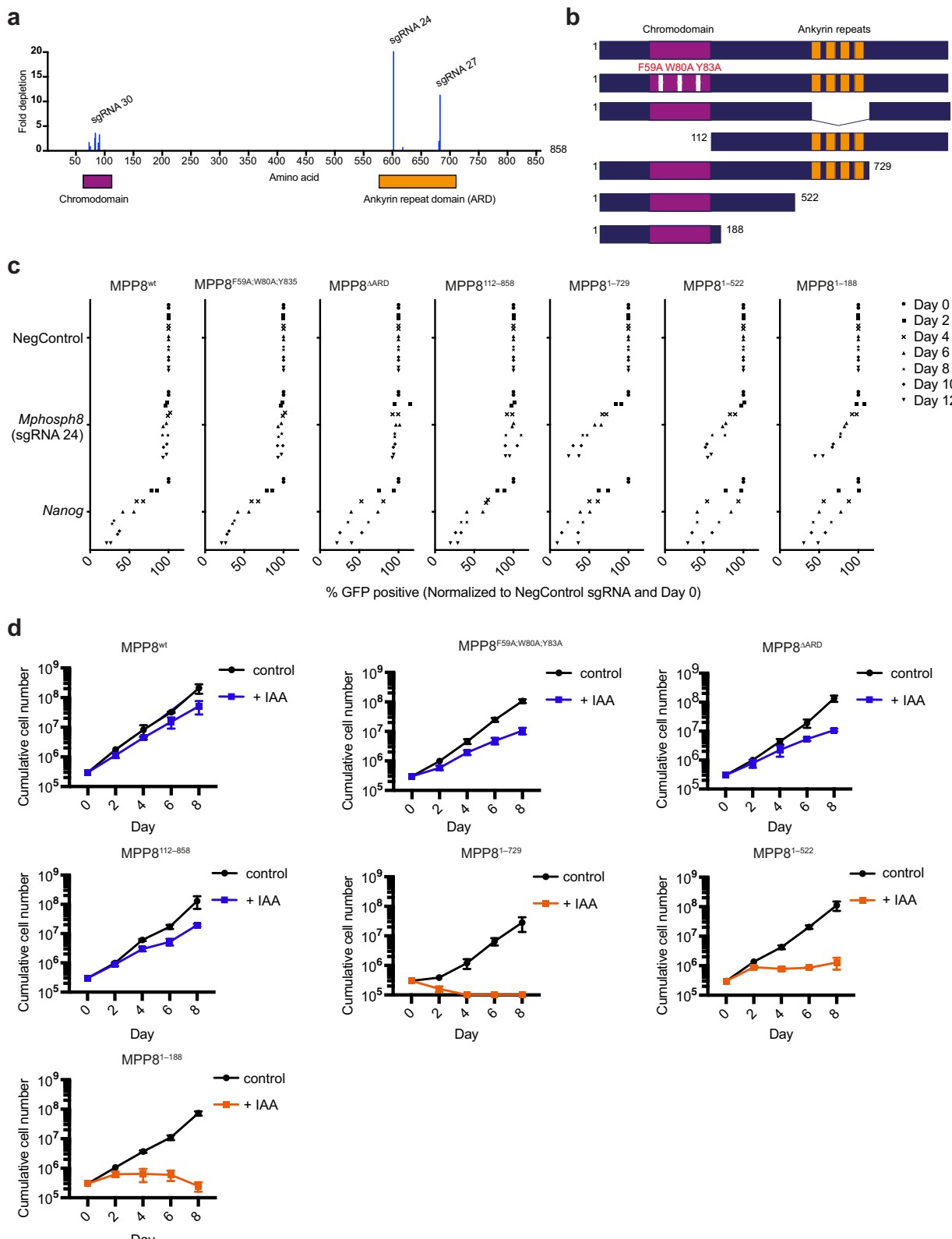

ChIP-qPCR of MPP8 in these cells showed that MPP8 retained its ability to bind to chromatin (Supplementary Fig. 6c), and that TASOR binding to chromatin is dependent on MPP8 (Supplementary Fig. 6d). This result is consistent with observations by others showing that the HUSH core complex is destabilized if one of its members is not expressed[15]. The requirement of MPP8 for the binding of TASOR to chromatin was further supported by the demonstration that TASOR was not found associated with its specific binding sites in mESCs expressing either the N- or C-terminus MPP8 mutants (Supplementary Fig. 6d).

Taken together, these results show that both MPP8 and TASOR require the chromodomain of MPP8 for their stable binding to chromatin, and that MPP8-mediated self-renewal of ground-state mESCs is independent of this stable association.

**Fig. 3 The essential function of MPP8 is independent of its chromo- and ankyrin-repeat domains but dependent on its very C-terminal region. a** Graph showing MPP8 amino acid sequence (X-axis) against the fold change reached for each sgRNA (Y-axis) used in the screen when comparing median normalized read counts from day one and day eleven. sgRNA 24 and 27 target the C-terminal ankyrin-repeat domain (ARD) while sgRNA 30 targets the N-terminal chromodomain. **b** Schematic representation of MPP8 proteins expressed in the rescue experiments. **c** Competition-based proliferation assays in mESCs stably expressing indicated *Mphosph8* cDNA resistant to targeting by sgRNA 24 as well as Cas9. Cells were further transduced with the respective sgRNA. GFP is monitored over a time course of 12 days. An sgRNA targeting the mESC-specific essential gene *Nanog* served as positive controls while a non-targeting sgRNA (NegControl) served as negative control. The percentage of GFP+ cells is normalized to the day zero measurement and the measurement of NegControl at the respective day ($n = 2$ independent experiments). **d** Growth curves of Mpp8$^{mAID}$ cell lines stably expressing MPP8 wild-type or the indicated MPP8 mutant proteins as well as OsTIR1 without or upon treatment with 500 μM IAA (mean ± s.d., $n = 3$ independent experiments). Source data are provided as a Source Data file.

**MPP8-mediated LINE1 control correlates with mESC maintenance.** To dissect the mechanism by which MPP8 promotes mESC self-renewal we focused our attention on transcriptional changes induced by MPP8 depletion. Consistent with previous findings[16,17], MPP8 was found to be enriched at transcriptionally active LINE element classes, such as L1Md_A and L1Md_T, indicating a potential direct role of MPP8 on LINE1 expression (Supplementary Fig. 7a). Hence, we investigated if loss of MPP8 would affect LINE1 expression.

Transcripts for the evolutionary young LINE1 A subclass as well transcripts encoding the L1ORF2 protein were significantly upregulated in MPP8-depleted mESCs, by more than 2-fold and 5-fold, respectively (Supplementary Fig. 7b). L1ORF2-encoding transcripts were upregulated more than 5-fold 48 h after induced proteolysis of MPP8, whereas SINE expression levels showed only modest changes (Fig. 5a). This observation was consistent with increased expression of the L1ORF1 protein after 72 h (Fig. 5b). In comparison, the levels of LINE1 expression were increased less than 1.5-fold in *Mphosph8* knockout MEFs (Supplementary Fig. 7c). To further understand if upregulation of LINE1 expression correlates with loss of functional MPP8, we assessed L1ORF2-encoding transcript levels in the presence of different MPP8 mutants. As expected, reintroduction of MPP8$^{wt}$ reduced L1orf2 transcript expression to baseline levels (Fig. 5c). Similarly, the expression of MPP8$^{112-858}$ was sufficient to maintain the repression of L1ORF2-encoding transcripts, while the expression of MPP8$^{1-729}$ was not (Fig. 5c).

To further investigate the role of MPP8 in regulating transcription, we determined the effect of MPP8 removal on genome-wide transcriptome level (Fig. 5d–g). We observed differential regulation of 722 transcripts as early as 6 h following MPP8 depletion, among these 396 arising from annotated genes and the remaining 326 from other annotated elements, such as repeats. 418 transcripts were upregulated, while 304 were downregulated and the majority of all differentially expressed transcripts was rescued by re-expression of wild-type MPP8 (609/722) (Supplementary Fig. 7d). The existence of downregulated transcripts could indicate the presence of both primary and secondary effects already at this early time point after MPP8 depletion. To understand which genes are potential direct targets of MPP8, we assigned the closest transcription start sites to each of the 396 differentially expressed genes ($|FC| > 1.5$, $p_{adj} <= 0.05$) or control genes ($p_{adj} > 0.05$) and computed the distance to high-confidence MPP8 binding sites (Supplementary Fig. 7e). Our analysis showed no significant difference in distance between differentially expressed and control genes, indicating no obvious direct regulatory role of MPP8 on differentially expressed genes observed at early time points of MPP8 depletion. In contrast, LINE1 elements, including the evolutionary young L1Md_Gf, L1Md_T, and L1Md_A elements, were significantly upregulated at early hours post depletion (Fig. 5d, Supplementary Fig. 7 f), and rescued by MPP8$^{wt}$ expression (Fig. 5e).

Moreover, the majority of high-confidence MPP8 DNA-binding sites directly overlapped LINE1 classes shown as

upregulated by RNA-seq: 25% of peaks overlapped evolutionary young LINE1 elements with strongest upregulation levels (FC > 2, $p_{adj} <= 0.05$), 24% overlapped older LINE1 classes found to be deregulated (FC > 1, $p_{adj} <= 0.05$) and 13% of peaks overlapped with both youngest and older classes (Supplementary Fig. 7g). Together, these observations support a direct role of MPP8 in repressing LINE1 expression.

Consistent with a potential key role of deregulated LINE1 expression in inhibiting mESC self-renewal, LINE1 elements remained de-repressed in MPP8-depleted mESCs expressing MPP8$^{1-729}$ (Fig. 5f and Supplementary Fig. 7h). Furthermore, expression of MPP8$^{112-858}$ was sufficient to rescue transcriptional changes, including evolutionary young LINE elements (Fig. 5g). RNA-seq analysis at 48 h post-auxin treatment showed differential regulation of 314 transcripts (Fig. 5h), 290 of which were re-repressed by wild-type MPP8, including four LINE1 classes (Fig. 5i, Supplementary Fig. 7 f). At this later time we observed more pronounced global transcriptional de-repression for MPP8$^{1-729}$-expressing cells (Fig. 5j and Supplementary Fig. 7i): 4481 transcripts were either up- or downregulated and overlapped the majority of transcripts deregulated in non-rescued cells (274/314). The greater number of deregulated transcripts and vast overlap between MPP8-depleted and cells expressing the N-terminal part of MPP8, potentially reflects secondary effects in relation to the observed cell death. Together with the severe proliferative defect of MPP8$^{1-729}$-expressing cells (Fig. 3d) and the observed destabilization of the HUSH core complex in presence of MPP8$^{1-729}$ (Supplementary Fig. 5b), the stronger transcriptional impact compared to deletion of MPP8 in wild-type cells might additionally suggest a dominant-negative role of C-terminally truncated MPP8. With exception of 17 transcripts, MPP8$^{112-858}$ expression efficiently repressed all transcriptional changes, including LINE1 expression (Fig. 5k), further supporting the hypothesis that HUSH-mediated LINE1 silencing is not connected to MPP8 binding to H3K9me3 through its chromodomain.

By profiling permissive chromatin post-translational modifications in the different cell lines using ChIP-seq, we found that H3K4me3 levels at LINE1 TSSs that were increased upon MPP8 depletion were efficiently rescued upon reintroduction of MPP8 wild-type as well as the N-terminal deletion mutant, while expression of the C-terminal deletion mutant did not rescue (Supplementary Fig. 8a). A similar change, although less strikingly, was also observed for H3K27ac levels (Supplementary Fig. 8b). We also found that LINE1 elements modulated by the loss of MPP8 were significantly longer compared to all LINE1 elements belonging to these classes (Supplementary Fig. 8c), while the contribution of young L1Md_T/A elements to both activated and all elements in these classes was similar (Supplementary Fig. 8d). Since evolutionary young, full-length LINE1 elements, which maintain intact promoter regions, are more likely to be transcribed compared to shorter elements that have often been rendered inactive through truncation[34], regulation by MPP8

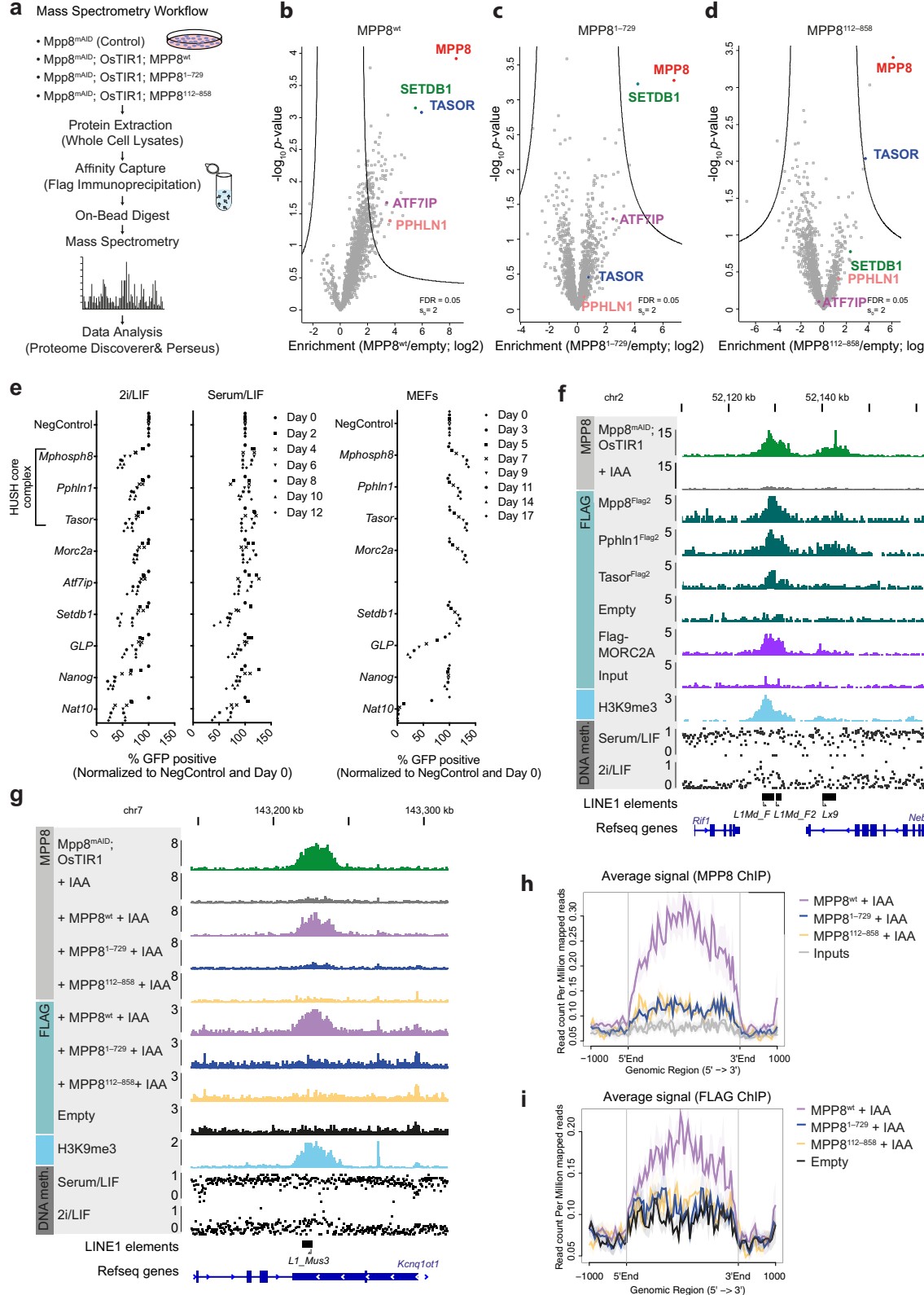

might be particularly required to safeguard the pluripotent epigenome at these transcription-permissive full-length LINE1 elements. Consistent with this hypothesis, we found young LINE1 elements to be de-repressed by MPP8 loss (Fig. 5h, Supplementary Fig. 7f). We conclude that MPP8 depletion or its C-terminal truncation in ground-state mESCs results in increased levels of permissive chromatin modifications at transcription start sites of

transcription-permissive LINE1 elements and their enhanced expression.

To further investigate the correlation between LINE1 expression and self-renewal capacity, we tested LINE1 expression levels of MPP8-depleted mESCs under different culture conditions. qPCR analysis showed increased levels of LINE1 expression in MPP8-depleted mESCs grown in 2i/LIF as compared to serum/

**Fig. 4 The essential function of MPP8 requires the formation of a stable HUSH core complex but is independent of its detectable chromatin binding. a** Schematic overview of the LC-MS/MS workflow for interactome analysis. **b–d** LC-MS/MS of Mpp8$^{mAID}$ cells expressing either ectopically double flag-tagged MPP8$^{wt}$, MPP8$^{1-729}$, or MPP8$^{112-858}$ as well as OsTIR1 treated with 500 µM IAA (12 h). Parental Mpp8$^{mAID}$ cells serve as background control ($n =$ 3 biologically independent samples for parental cells, MPP8$^{wt}$ and MPP8$^{1-729}$, $n = 2$ for MPP8$^{112-858}$). HUSH complex members are color-highlighted. **e** Competition-based proliferation assays indicated Cas9-expressing cell lines. An sgRNA targeting the core essential gene *Nat10* served as positive control while a non-targeting sgRNA (NegControl) served as negative control. An sgRNA against *Nanog* served as mESC-specific positive control. The percentage of GFP+ cells was normalized to the day 0 measurement and the measurement of NegControl at the respective day (mESCs: $n = 2$ independent experiments, MEFs: $n = 1$). **f, g** Representative genome browser tracks showing: (**f**) MPP8 ChIP-seq in Mpp8$^{mAID}$; OsTIR1 cells ± 500 µM IAA (16 h); FLAG ChIP-seq of endogenously FLAG-tagged *MPP8*, *PPHLN1* and *TASOR* proteins in 2i/LIF-cultered mESCs (parental untagged mESCs served as empty control) or overexpressed FLAG-MORC2A in serum/LIF-cultured mESCs (input served as control) (data taken from Ref. [32]); (**g**) MPP8 ChIP-seq in indicated Mpp8$^{mAID}$; OsTIR1 cell lines ± 500 µM IAA (16 h), FLAG ChIP-seq in indicated Mpp8$^{mAID}$; OsTIR1 cell lines + 500 µM IAA (16 h) (E14 cells served as empty control) (**f, g**) H3K9me3 ChIP-seq signal in Mpp8$^{mAID}$; OsTIR1 cells; DNA methylation profiles in ESCs grown under serum/LIF or 2i/LIF culture conditions (0 = unmethylated, 1 = fully methylated CpG sites; data taken from Ref. [35]). Repeatmasker track showing the location of relevant LINE1 elements is indicated at the bottom. **h, i** Aggregate plot comparing the average MPP8 (**h**) and FLAG (**i**) ChIP signal, respectively, over high-confidence MPP8 peaks ($n = 55$) in indicated Mpp8$^{mAID}$; OsTIR1 cell lines + 500 µM IAA (16 h). Input signal (**h**) and empty parental cells (**i**), respectively, served as control. Source data are provided as a Source Data file.

---

LIF, as exemplified by a 3-fold higher level of L1orf2 in 2i/LIF versus serum/LIF. (Supplementary Fig. 9a). This is consistent with previous observations showing more pronounced LINE1 reactivation in ground-state compared to metastable mESCs upon shRNA-mediated *Mphosph8* depletion[17]. Moreover, while the addition of the GSK inhibitor to serum/LIF culture did not increase L1orf2 expression in MPP8-depleted cells further, serum/LIF culture supplemented with MEKi or 2i, for which we observed lower proliferation capacity (Supplementary Fig. 3d), yielded increased L1orf2 expression levels following MPP8 removal (Supplementary Fig. 9b). These results are in line with the hypothesis that mESC self-renewal is impaired when LINE1 expression exceeds a certain level.

Since DNA methylation is increased in mESCs grown in serum/LIF as compared to mESCs in 2i/LIF[35], we hypothesized that the increased level of DNA methylation could be an additional mechanism to repress LINE1 expression and hence serum/LIF-grown mESCs become independent of MPP8 for their self-renewal. In contrast to *Dnmt* wild-type ESCs, targeting of *Mphosph8* in mESCs knockout for *Dnmt3a*, *Dnmt3b*, and *Dnmt1* (*Dnmt* tKOs)[36] showed a similar increase of LINE1 transcription independently of the culture conditions (Supplementary Fig. 9c). These results suggest that LINE1 elements in metastable mESCs are repressed by two independent mechanisms involving DNA methylation and MPP8, respectively. In agreement with these results, targeting of MPP8 under serum/LIF culture conditions, as expected, barely impacted cell growth whereas MPP8 was essential for *Dnmt* tKO cells grown in serum/LIF (Supplementary Fig. 9d). Taken together, these results show that DNA methylation and MPP8 independently silence LINE1 expression in metastable mESCs, and that inactivation of both of these repressive mechanisms leads to growth arrest.

In summary, our results show a strong correlation between increased levels of LINE1 expression and loss of mESC self-renewal induced by MPP8 degradation, suggesting that the increased LINE1 expression could be causing the phenotype.

**H3K9me3 is dispensable for maintained LINE silencing by MPP8.** To test the requirement of H3K9me3 both for the recruitment of MPP8 and for HUSH-mediated transcriptional repression, we performed MPP8 ChIP in mESCs lacking the H3K9me1/2 methyltransferases G9a/GLP[37], the H3K9me1/2/3 methyltransferases SUV39H1/H2[38] and in cells conditionally depleted for H3K9me3 methyltransferase SETDB1[39] (Fig. 6a, b). The loss of G9a/GLP impacted MPP8 recruitment (Fig. 6a) at some of the investigated MPP8-bound regions, in agreement with the proposed role of G9a in the recruitment of MPP8 via

methylation of ATF7IP[14]. While SUV39H1/H2-mediated H3K9me3 has been shown to contribute to silencing of evolutionary young LINE1 elements in serum/LIF-grown mESCs[40], we did not find evidence for a role of SUV39H1/H2 at MPP8-bound regions, as neither H3K9me3 levels (Fig. 6b) nor MPP8 recruitment (Fig. 6a) were altered in *Suv39h1/h2* dKOs at the investigated target loci. Consistent with a described role for SETDB1 in MPP8 chromatin binding[15,41], its deletion led to loss of MPP8 recruitment to all investigated target loci (Fig. 6a and Supplementary Fig. 10a) concomitant with H3K9me3 reduction (Fig. 6b). Therefore, we conclude that SETDB1 is both the main H3K9me3 methyltransferase responsible for H3K9me3 deposition as well as MPP8 recruitment at MPP8-bound regions in the ground-state mESCs.

To determine the role of H3K9me3 in the maintenance of HUSH-mediated LINE1 silencing, we performed H3K9me3 ChIP-seq after MPP8 degradation with or without the presence of exogenously expressed wild-type or mutant MPP8 proteins. After 6 h of addition of auxin, no H3K9me3 changes were observed in any of the conditions (Fig. 6c and Supplementary Fig. 10b, c; 11a–c). After 48 h of MPP8 removal, MPP8 target loci showed a reduction in H3K9me3 in MPP8-depleted cells, consistent with published reports of H3K9me3 loss in *Mphosph8* knockout cells[15,16] (Fig. 6d and Supplementary Fig. 10b, d; 11d–f). Moreover, both cell lines expressing N-terminally or C-terminally truncated MPP8 versions failed to maintain H3K9me3 at MPP8 target sites. Strikingly, since the N-terminally truncated MPP8 maintains the repression of LINE1 elements, these results demonstrate that H3K9me3 is not required for MPP8 to maintain the repression of these repeats.

## Discussion

By using an improved CRISPR/Cas9 screening approach targeting functional domains, we have identified 146 proteins with a potential role in regulating epigenetic features contributing to the self-renewal of mESCs cultured in 2i/LIF. We demonstrate that MPP8 is vital for the self-renewal of ground-state pluripotent stem cells by maintaining cell cycle progression and protecting from spontaneous differentiation while depletion of MPP8 in metastable mESC cultures only mildly affects proliferation. The latter observation is in line with a recent study that established stable knockout *Mphosph8* mESCs in serum/LIF[14]. Our results further show that the two other HUSH core complex members, PPHLN1 and TASOR, are equally essential for ground-state mESCs. Consistent with our in vitro observations, in vivo, both *Pphln1*$^{-/-}$ and *Tasor*$^{-/-}$ embryos show early peri-gastrulation lethality around embryonic day 7.5 (E7.5)[42–44]. Additionally, our

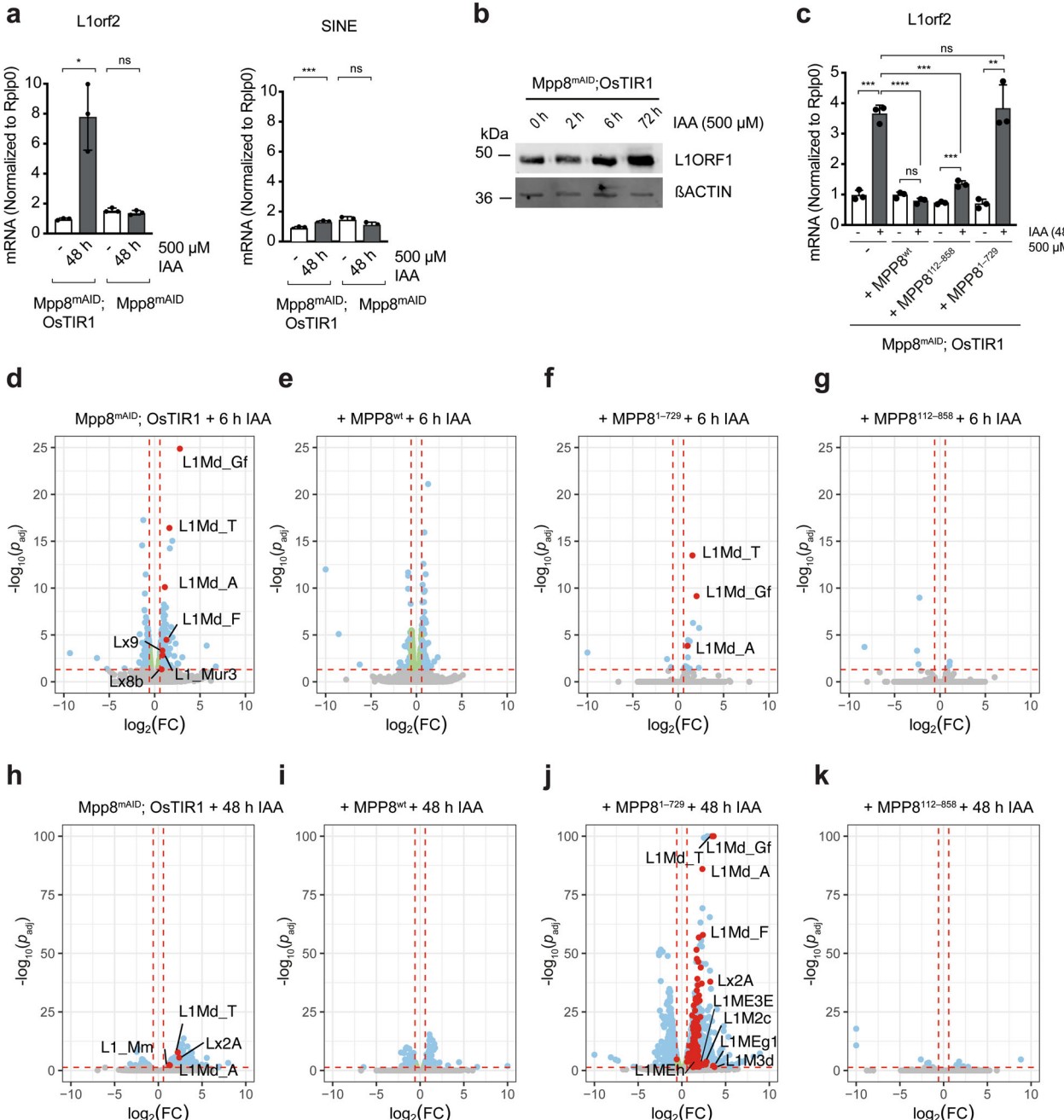

**Fig. 5 MPP8 degradation leads to increased LINE1 expression. a** qRT-PCR analysis of transcripts encoding LINE1-encoded ORF2 protein or SINEs in Mpp8$^{mAID}$; OsTIR1 or Mpp8$^{mAID}$ (control) cells grown in 2i/LIF treated with 500 µm IAA for 48 h (mean ± s.d., n = 3 independent experiments). p = 0.0348 (L1orf2) and p = 0.0009 (SINE) for Mpp8$^{mAID}$; OsTIR1 ± IAA and p = 0.3068 (L1orf2) and p = 0.1988 (SINE) for Mpp8$^{mAID}$ ± IAA (two-tailed paired Student's t test). **b** L1ORF1 protein expression in Mpp8$^{mAID}$; OsTIR1 cell line grown in 2i/LIF upon IAA treatment and βACTIN (loading control) (n = 1). **c** qRT-PCR analysis of L1orf2 transcript, encoding LINE1-encoded ORF2 protein, transcript expression in Mpp8$^{mAID}$; OsTIR1 cells grown in 2i/LIF stably expressing MPP8 wild-type, indicated MPP8 mutant, or no additional cDNA treated with 500 µm IAA for 48 h (mean ± s.d., n = 3 independent experiments). p = 0.0093 (−), p = 0.1973 (wt), p = 0.0089 (112–858) and p = 0.0139 (1–729) for indicated cells ± IAA (two-tailed, paired Student's t test) and p = 7.900E-05 (wt), p = 0.0002 (112–858) and p = 0.7296 (1–729) comparing the indicated auxin-treated cell line to auxin-treated Mpp8$^{mAID}$; OsTIR1 cells (two-tailed, unpaired Student's t test). **d**–**k** Volcano plots presenting differentially regulated transcripts upon treatment with 500 µM IAA for 6 h (**d**–**g**) or 48 h (**h**–**k**) in Mpp8$^{mAID}$; OsTIR1 cells grown in 2i/LIF stably expressing MPP8 wild-type and indicated MPP8 mutants as assessed by RNA-seq. Dashed red lines: $p_{adj} < 0.05$, |FC| > 1.5. Red dots: significantly changed LINE1 classes. Blue dots: Significantly changed transcripts (p < 0.05, |FC| > 1.5). Green dots: Significant changes with |FC| < 1.5. Gray dots: Not significant (n = 2 biologically independent samples). In (**j**) the top ten differentially expressed LINE1 classes based on fold change are highlighted. For visualization purposes, transcripts surpassing |log$_2$FC| > 10 or −log$_{10}$($p_{adj}$) > 100 are plotted as |log$_2$FC| = 10 or −log$_{10}$($p_{adj}$) = 100, respectively. *p < 0.05, **p < 0.01, ***p < 0.001, ****p < 0.0001, ns = not significant, FC = fold change. Source data are provided as a Source Data file.

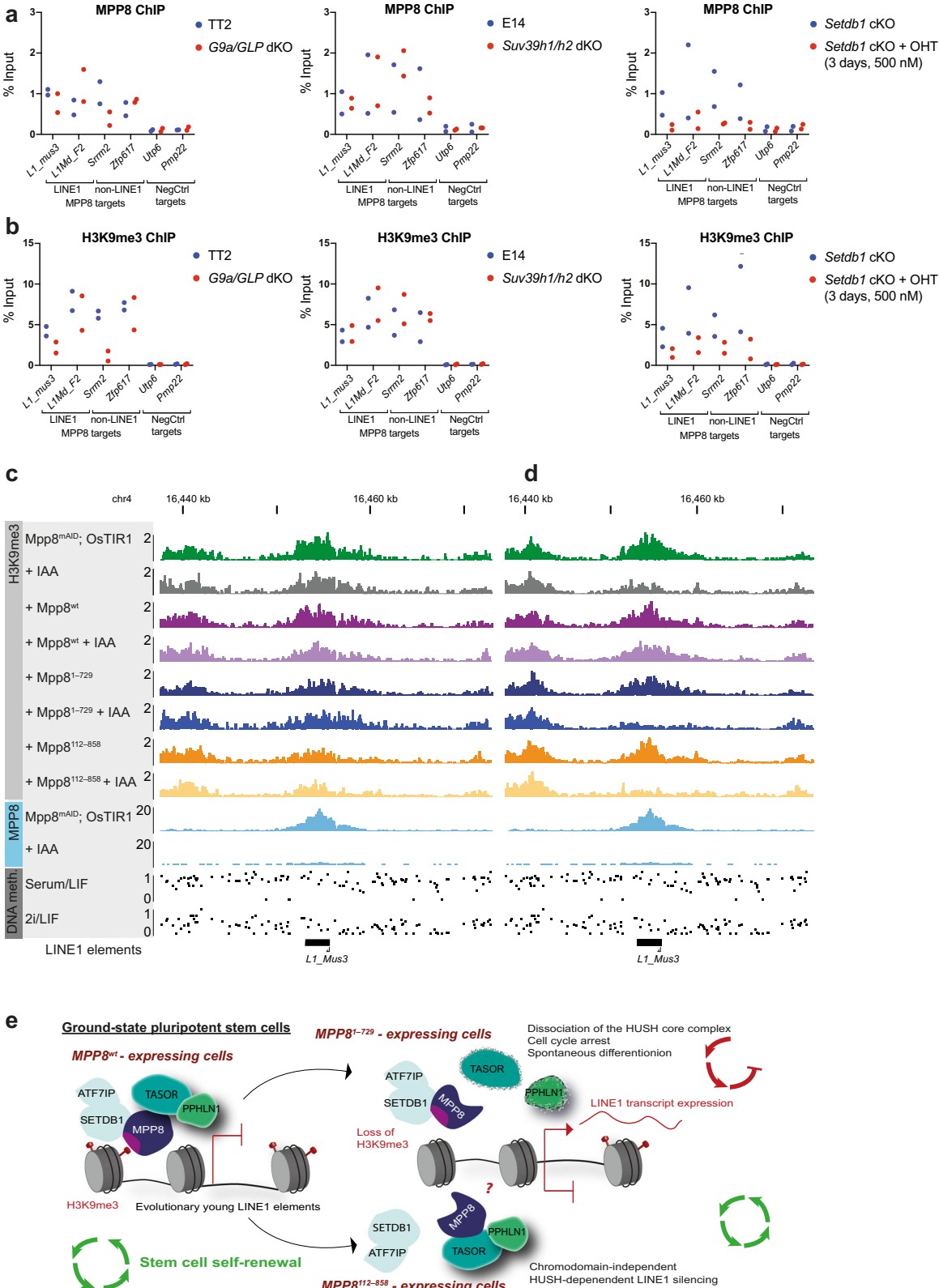

results suggest that there is a switch between the naïve pluripotency state and the more differentiated embryonic stages after which the HUSH complex is no longer essential.

The lethal phenotype induced by the inactivation of MPP8 in ground-state pluripotent cells correlates with the increased expression levels of LINE1 elements. This may reflect a causal role of unphysiologically increased LINE1 expression levels leading to

cell death in ground-state mESCs. LINE1 element expression is thought to require tight regulation as increased activity was suggested to contribute to genome instability and altered gene transcription[45]. In vivo studies in early developmental murine stages, in which LINE1 activation peaks at 2-cell stage and can be readily detected at 8- and 16-cell stage, have demonstrated that prolonged activation of LINE1 transcription beyond the 2-cell

**Fig. 6 SETDB1 is required for MPP8 recruitment in mESCs, while H3K9me3 is dispensable for maintaining LINE1 repression. a, b** MPP8 (**a**) and H3K9me3 (**b**) ChIP-qPCR at two MPP8 binding sites that overlap LINE1 elements (*L1_mus3, Kcnq1ot1* locus), *L1Md_F2*, chr. 2), two non-LINE1 MPP8 binding sites (*Srrm2, Zfp617*) and two negative control (NegCtrl) loci not bound by MPP8 (*Utp6, Pmp22*) in indicated cell lines grown in 2i/LIF. E14 and TT2 cells served as parental control for *Suv39h1/h2* dKOs and *G9a/Glp* dKOs, respectively. Conditional *Setdb1* KO (Setdb1 cKO) cells were treated with 500 nM 4-hydroxytamoxifen (OHT) for 3 days (*n* = 2 independent experiments). **c, d** Representative genome browser tracks of H3K9me3 ChIP-seq in indicated Mpp8$^{mAID}$; OsTIR1 cell lines grown in 2i/LIF ± 500 µM IAA for 6 h (**c**) or 48 h (**d**); MPP8 ChIP-seq in Mpp8$^{mAID}$; OsTIR1 cells ± 500 µM IAA for 16 h; DNA methylation profiles in ESCs grown under serum/LIF or 2i/LIF culture conditions (0 = unmethylated, 1 = fully methylated CpG sites; data taken from Habibi et al.[35]); Repeatmasker track showing the location of relevant LINE1 elements. **e** Model for MPP8-mediated maintenance of stem cell self-renewal in ground-state pluripotent stem cells. In wild-type MPP8-expressing mESCs MPP8 is recruited to chromatin by SETDB1. MPP8 binds to TASOR through its C-terminal part, enabling the integrity of the core complex, and to chromatin and SETDB1 through the N-terminal part containing the chromodomain. This configuration allows maintenance of LINE1 repression and hence stem cell self-renewal of mESCs. Removal of the C-terminal part evokes destabilization of the complex as binding of MPP8 to TASOR is lost, leading to de-repression of evolutionary young LINE1 elements and loss of stem cell identity. In cells that lack MPP8's N-terminal part stable binding of MPP8 and TASOR is lost from chromatin. MPP8 can no longer bind to SETDB1 and maintenance of H3K9me3 is lost. However, the intact HUSH core complex continuously represses the LINE1 elements and support self-renewal of mESCs, demonstrating that stable association with chromatin and maintenance of H3K9me3 are not required. Source data are provided as a Source Data file.

stage or their ablation leads to impaired preimplantation development[46]. In addition, another recent study suggested that downregulation of LINE1 transcriptional levels using shRNAs and antisense oligos likewise causes loss of stem cell self-renewal[47]. Taken together with our data, this suggests that LINE1 expression levels have to be precisely regulated in mESCs. Our data show that MPP8 and the other HUSH core complex members are essential for this regulation. Moreover, we show that MPP8 is required for the self-renewal of ground-state mESCs and suggest that the loss of self-renewal upon loss of MPP8 is caused by an increase in LINE1 expression. This hypothesis is in agreement with recent reports demonstrating that ectopic LINE1 overexpression leads to G1 arrest in RPE cells[48].

In committed somatic cells, DNA methylation has a crucial role in controlling LINE1 expression[49]. Activation of LINE1 elements after fertilization is associated with loss of DNA methylation, which is re-established during blastocyst development[50]. We show that mESCs grown in serum/LIF and MEFs, in which LINE1 elements display substantial levels of DNA methylation (32-50% and >80%, respectively[49]), do not require MPP8, and that knockout of MPP8 only leads to a slight increase in LINE1 expression. mESCs grown in 2i/LIF, in which LINE1 DNA methylation levels are reduced to <20%[35], die in the absence of MPP8. Hence, these results imply that the HUSH complex is essential to safeguard the genome from aberrant LINE1 activation in cells that display low levels of DNA methylation. This observation also raises the exciting possibility that the HUSH complex displays an essential regulatory role in LINE1 control in other hypomethylated cell types, such as germ cells and cancer cells[51]. Interestingly, deletion of *Setdb1* in early germ cells[52] was shown to have little impact on reactivation of LINE1 elements besides low levels of DNA methylation and a clear decrease of H3K9me3 at specific H3K9me3-enriched LINE1 loci. Similarly, no LINE1 de-repression was observed upon removal of *Setdb1* in growing oocytes[53], suggesting that if HUSH is required for the maintenance of LINE1 repression in these hypomethylated settings, it would be independent of SETDB1.

In current models, H3K9me3 is believed to play a key role for both recruitment and maintenance of HUSH-mediated LINE1 silencing[15,33]: MPP8 is recruited to H3K9me3-positive regions via its chromodomain, which is followed by the binding of the HUSH core complex and the recruitment of SETDB1. This leads to the further spreading of H3K9me3 and transcriptional repression of LINE1 elements. Our results provide new mechanistic insights into how MPP8 and the HUSH complex regulate LINE1 expression. We show that the C-terminus of MPP8 is essential for its biological function, and that it is required for the interaction with TASOR and for the stability of the HUSH

core complex. In contrast, the N-terminal region of MPP8, containing the chromodomain, is required for its interaction with SETDB1.

To our surprise, the N-terminus of MPP8, which is required for the interaction with SETDB1, H3K9me3 binding and detectable binding to HUSH target genes, is not required for the biological function of MPP8 in maintaining self-renewal of ground-state pluripotent mESCs. This observation appears in conflict with the described essential role of SETDB1, which we also confirm in our study, and H3K9me3 in repressing LINE1 element expression. However, our results demonstrate that an MPP8 mutant, lacking the N-terminus, maintains LINE1 element expression at physiological levels. Moreover, we show that both the chromodomain-containing N-terminus and the C-terminus of MPP8 are required for detectable binding to HUSH target loci. Therefore, stable binding of MPP8 to chromatin may both require the interaction between the chromodomain of MPP8 and H3K9me3 and the stabilization of this binding via the C-terminal of MPP8 with other components of the HUSH complex. Thus, capturing a less stable chromatin binding of the MPP8$^{112-858}$ or MPP8$^{1-729}$ mutants may be below the sensitivity limit of ChIP experiments. Alternatively or in addition, the intrinsic nature of these interactions could be more transient: recent work has suggested an additional role for interactions caused by phase separation in HP1α-mediated gene silencing[54]. MPP8, similarly to HP1α, contains an intrinsically disordered region (IDR) downstream of its chromodomain (MPP8$^{109-529}$, as determined by protein intrinsic disorder predictor PONDR® VSL2[55]), which could potentially contribute to retained chromatin sequestration in MPP8$^{112-858}$ mutants. Yet, it is important to emphasize that the stable chromatin binding is not required for the essential function of MPP8 in ground-state mESCs. When our manuscript was under review, another study was published addressing the functional requirement of MPP8 domains in HUSH-mediated repression of transgenes[56]. Consistent with our structure-function analysis addressing the requirement of MPP8's domains for endogenous LINE1 element repression and stem cell self-renewal, genetic complementation of *Mphosph8* knockout cells with the human MPP8$^{500-860}$ protein was found to silence a GFP reporter construct, whereas MPP8$^{1-728}$ did not. Hence, the requirement of MPP8's C-terminal region is conserved between both its silencing activity on endogenous LINE1 elements and that on transgenic insertions, while the chromodomain is dispensable for both functions.

Taken together, our findings provide evidence for an alternative mechanism (Fig. 6e), in which SETDB1 and H3K9 methylation are required for setting up transcriptional repression and recruitment of MPP8; once initial repression is established,

the presence of a stable HUSH core complex is sufficient to maintain transcriptional repression while its interaction with H3K9me3 and stable binding to chromatin are no longer needed.

## Methods

**Cell culture.** An mESC line derivative of feeder-independent E14-TG2a parental cells (129/Ola strain) was used for all experiments with exception of experiments using Mpp8[Flag2], Tasor[Flag2], and Pphln1[Flag2] cell lines which were derived from TCF2.2 (F1 hybrid 129S6;C57BL/6N lines) cells. mESCs were grown in 2i/LIF medium as described[57]. For indicated experiments, mESCs were grown in serum/LIF medium (Glasgow Minimum Essential Media (GMEM), 10% heat-inactivated Fetal Bovine Serum (HI-FBS), 1% Penicillin/Streptomycin, 2 mM Glutamax (Gibco), 100 μM β-Mercaptoethanol (Gibco), 1 mM Sodium Pyruvate (Gibco), 0.1 mM MEM Non-Essential Amino Acids Solution (MEM-NEAS, Gibco) and LIF (made in-house). NSCs were cultured as described[58]. HEK293FT cells and Trp53$^{-/-}$ MEFs were cultured in Dulbecco's Modified Eagle Medium (DMEM, Gibco) supplemented with 10% heat-inactivated Fetal Bovine Serum and 1% Penicillin/Streptomycin. NIH 3T3 mouse embryonic fibroblasts were maintained in DMEM (Gibco) supplemented with 10% Fetal Bovine Serum, 2 mM Glutamax (Gibco), 1% Penicillin/Streptomycin (Gibco) and 1 mM Na$_2$SeO$_3$.

**Generation of mESC knockin cell lines.** For the generation of the endogenous auxin-inducible degradation system for MPP8 sgRNAs targeting the stop codon region were cloned into eSpCas9(1.1)-T2A-Puro (a kind gift from Ian Chambers). Left and right homology arms, as well as the mAID-T2A-BFP middle part were ligated into a modified pUC19 vector backbone (a kind gift from Steven Pollard). mESCs were co-transfected with sgRNA- and donor-vector using Lipofectamine 3000 and sorted 48 h later for GFP/BFP double-positive cells. Homozygous clones were transfected with pPB-OsTIR1-P2A-mCherry and selected with hygromycin B (100 μg/mL). Oligos are listed in Supplementary Table 2.

For the generation of endogenous 2xFlag tagged lines, sgRNAs targeting the stop codon region of the respective gene were cloned into PX458 (Addgene plasmid # 48138). A donor 2xFlag tag was co-transfected together with the sgRNA plasmid. GFP-positive cells were single-cell sorted and colonies were genotyped by PCRs. Oligos are listed in Supplementary Table 3.

For the generation of Mphosph8 knockout NIH3T3, cells were transfected with PX458 vector containing a sgRNA targeting the Mphosph8 genomic locus and cells were single cell sorted for GFP.

For the establishment of stably Cas9-expressing mESCs and MEFs a Lenti-EF1a-Cas9-2A-Blast construct (a kind gift from Feng Zhang, Addgene: 52962) was transduced and selected with blasticidin (10 μg/ml).

Homozygous insertions, knockouts and stable expressions were confirmed through Sanger sequencing and western blot.

**sgRNA vector generation.** The U6-sgRNA-SFFV-puro-P2A-EGFP vector for the epi-sgRNA-library and competition-based proliferation assays was cloned as follows: the pL-CRISPR.SFFV.GFP vector, a kind gift from Benjamin Ebert (Addgene: #57827), was digested with NheI and BamHI, dephosphorylated using Calf Intestinal Alkaline Phosphatase (NEB) and purified using the Qiagen PCR Purification kit. Puromycin resistance was amplified from a lentiGuide-Puro vector (a kind gift from Feng Zhang, Addgene # 52963). The pieces were ligated using the Quick Ligaton™ kit (NEB) (Primers are listed in Supplementary Table 4).

**Virus production and lentiviral transduction.** HEK293FT cells were plated 24 h prior to co-transfection of target vector, psPAX2 (Addgene #12260) and pCMV-VSV using a standard calcium phosphate protocol. The next day, the medium was changed. 48 h later the virus was harvested, and target cells were transduced. The following day the medium was changed and selection with the respective antibiotics was applied.

**Design of the domain-focused CRISPR/Cas9 mouse epi library.** Mouse (mm10) PFAM protein domain annotations were retrieved from the UCSC genome annotation database, converted to BED6 format using BEDTools[59] and intersected with transcript annotations using BEDTools intersectBed. FASTA sequences were extracted using BEDTools fastaFromBed.

To extract gRNAs from the FASTA sequences, low complexity regions were trimmed off, and PAM (GG) motif positions were searched and a region of 30 bp was extracted. On-target efficiency was predicted using Azimuth[60]. To identify potential off-targets, 23 bp gRNAs were extracted from the 30 bp and aligned to mouse whole-genome index using bowtie[61], allowing up to three mismatches per end-to-end alignment and with a maximum of 35 alignments per gRNA (bowtie arguments: "-v 3 -k 35"). These alignments were parsed to remove any gRNA with more than 1 exact match (or with multiple identical regions) in the genome or with more than 25 alignments (to exclude gRNAs with large number of off-targets). BEDTools intersetBed utility and UCSC genome annotations (in GTF format) were used to group the remaining set of alignment to off-targets in exons, introns, and intragenic regions. In the same step, CFD score[60] was used to predict the off-target affinity of the gRNA to off-target sequences retrieved from alignment.

For each gene, guides were ranked for the following: Azimuth on-target efficiency score (descending order), off-targets mapping to exons of the genome, maximum off-target score for exonic off-targets, off-targets mapping to introns of the genome, maximum off-target score for intronic off-targets, off-targets mapping to intragenic regions of the genome and maximum off-target score for intragenic off-targets (increasing order). For each guide, a rank score was calculated as the mean of the ranks. For each gene, top scoring guides were defined as guides with the lowest rank scores.

To select gRNAs a domain ranking strategy was used. Protein domains were ranked based on the perceived order of importance. All gRNA sequences for a gene were ranked based on the domain annotated to the guide region and 10–11 best were picked.

**Array oligo synthesis and library amplification.** The oligonucleotide library consisting of 12472 unique sequences (Supplementary Data 1) was synthesized by CustomArray. Each full-length oligonucleotide (60 nt) consists of a unique sgRNA sequence of 20 nt length starting with a G, flanked by 20 nt on either site containing a recognition sequence for the BsmBI restriction enzyme. The library was amplified by PCR using Phusion® High Fidelity DNA Polymerase (NEB) using 20 cycles of 98 °C, 10 s; 61 °C, 30 s; 72 °C, 10 s (Supplementary Table 4). PCR products were size-selected and further purified using the Qiagen PCR Purification kit.

**Pooled library cloning.** The amplified library was cloned into the target vector using a 2 in 1 digest and ligation protocol: 0.92 ng of amplified library and 50 ng of target vector were mixed 1 μl of NEB buffer 3, 1 μl of 10 mM ATP, 0.5 μl BsmBI restriction enzyme (NEB) and 0.5 μl T4 ligase (NEB). 12 cycles of alternating 5 min at 42 °C and 5 min at 16 °C were followed by 15 min at 55 °C. To maintain the library representation 30 such ligation reactions were performed and after transformation in Stellar™ competent cells (Clontech) colony numbers were compared to a negative control reaction (not containing amplified library pool) to assess cloning efficacy, yielding a library representation of 673X. The colonies were scraped off the plates, pooled and plasmids were purified using NucleoBond® Xtra Maxi kit (Macherey-Nagel).

**Negative selection screen in mESCs with mouse epi library.** Virus containing the epi library was produced as described above. $48 \times 10^6$ cells were transduced at a MOI of approximately 0.3 to achieve a library representation of 1000 X. Cells were selected with puromycin (1 μg/ml) throughout the screen. Three days post-transduction, cells were analyzed for GFP (BD Bioscience LSRFortessa) and a reference sample (Day 1) was taken. Cells were further passaged, and cells were taken seven (Day 7) and eleven (Day 11) days later. The procedure was repeated in two biologically independent rounds.

**High-throughput sequencing of the CRISPR/Cas9 screen.** Genomic DNA was extracted using the DNeasy Blood & Tissue kit (Qiagen). Amplification was performed in multiple PCR reactions each containing 5 μg gDNA, 3 μl of 10 μM forward and reverse primer and 50 μl of 2X KAPA HiFi Hotstart ready mix (Roche) for 20 cycles of 98 °C, 20 s; 60 °C, 15 s; 72 °C, 30 s. To barcode samples all PCRs were pooled and for each sample three reactions containing 10 μl of the first PCR, 1 μl of each F/R Illumina primer as well as 50 μl of 2X KAPA HiFi Hotstart ready mix (Roche) were further amplified using 20 cycles of 98 °C, 20 s; 60 °C, 15 s; 72 °C, 30 s. The PCR products were extracted from a 1% agarose gel using the QIAquick Gel Extraction Kit (Qiagen) and further purified using the Qiagen PCR Purification kit. The libraries were quantified using the KAPA Library Quantification Kit (Roche, KR0405), pooled in equimolar quantities and sequenced on an Illumina NextSeq 550 (75 bp single-end). Primer sequences are listed in Supplementary Table 4.

**Analysis of the CRISPR/Cas9 screen.** Reads were mapped against the sgRNA library with bowtie[61] (-m 1 -v 1). The read counts for each library were first normalized by the library's read count mean to account for differences in sequencing depth between libraries. Then, sgRNAs that were present with read values falling into the lowest 2 percentile in day 0 libraries were removed. Normalized read count values were first averaged between the two biologically independent replicates and the fold change between day 7 and day 1 or day 11 and day 1, respectively, was determined.

**Competition-based proliferation assay.** sgRNAs were designed using the CRISPR design tool (https://portals.broadinstitute.org/gpp/public/analysis-tools/sgrna-design) from the Broad Institute (Supplementary Data 1, Supplementary Table 5). Each sgRNA was cloned into the U6-sgRNA-SFFV-puro-P2A-EGFP vector.

Cells expressing Cas9 were transduced with the respective sgRNA. 24 h later the media was changed, and cells were briefly selected with puromycin (1 μg/mL). Two days after transduction the cells were analyzed for GFP expression by flow cytometry and re-plated with blasticidin (10 μg/mL). At indicated time points, cells were again analyzed by flow cytometry, and re-seeded. At day 2, an aliquot was taken for extraction of genomic DNA and analysis by TIDE (https://tide.nki.nl/)

(Supplementary Table 6). The gating strategy is exemplified in Supplementary Fig. 12a.

**Rescue experiments with ectopic expression of MPP8.** *Mphosph8* cDNA was purchased from Dharmacon (cloneID 40131003), an N-terminal Kozak sequence and C-terminal double-flag tag were added through PCR amplification, and subsequently cloned into a piggyBAC backbone. The cDNA was rendered resistant to targeting by sgRNA *Mphosph8* 24 by inserting a silent mutation in the PAM using the Q5 site directed mutagenesis kit (NEB). The primers used were designed by NEBaseChanger[TM] (NEB) (Supplementary Table 7). For rescue experiments, cells were transfected with a 2:1:1 ratio of pPB-MPP8_Flag2-Blast, pPB-Cas9-2A-mCherry-Hygro and pBase constructs using Lipofectamine 3000 (Thermo scientific) according to the manufacturers' instructions. For rescue experiments using expression into the conditional Mpp8[mAID] depletion system, cells were transfected with a 2:1:1 ratio of pPB-MPP8_Flag2-Blast, pPB-OsTIR1_HA-P2A-mCherry-Hygro and pBase. Pools were selected with blasticidin (10 µg/mL) and hygromycin B (100 µg/mL) for approximately 10 days.

**Growth curves.** $3 \times 10^5$ cells were plated in triplicates in six-well plates under the respective culture conditions. Every 48 h, cells were split, counted using a Neubauer counting chamber and $3 \times 10^5$ cells were re-seeded.

**Alkaline phosphatase staining.** $1 \times 10^3$ mESCs were seeded into a six-well plate and cultured in the respective media for 5–7 days. Cells were fixed in citrate-acetone-formaldehyde and stained using the Leukocyte Alkaline Phosphatase Kit (Sigma) according to the manufacturer's instructions.

**Cell cycle analysis.** Approximately $10 \times 10^6$ cultured cells were treated with 20 µM EdU and incubated for 10 min at 37 °C. Cells were harvested and fixed in ice-cold 96% EtOH. Fixed cells were permeabilized and stained with fluorescent dye acid using the Click-iT Edu assay Kit (Life Technologies) according to the manufacturer's instructions. Following staining, cells were washed and resuspended in PBS containing propidium iodide and analyzed by flow cytometry (BD Bioscience LSR II) using BD FACSDiva v.8.0.2 (BD Bioscience) for the acquisition and subsequent data analysis using FlowJo v.9/10 (FlowJo LLC). Statistical analysis was performed using GraphPad Prism v.8.4.2 (GraphPad Software, LLC). The gating strategy is exemplified in Supplementary Fig. 12b.

**RNA isolation, cDNA synthesis, qRT-PCR.** Total RNA was isolated using the RNeasy mini kit (QIAGEN) according to the manufacturer's instructions. One microgram was reverse transcribed using the Transcriptor Universal cDNA Master (Roche). Quantitative PCR with reverse transcription (qRT-PCR) was performed in technical triplicates using LightCycler 480 SYBR Green I Master (Roche) on a LightCycler 480 Instrument II (Roche). Primer sequences are listed in Supplementary Table 8. Statistical analysis was performed using GraphPad Prism v.8.4.2 (GraphPad Software, LLC).

**Processing and analysis of RNA-seq data.** Total RNA was isolated using the RNeasy kit (74104, Qiagen) and RNA quality was assessed using the RNA 600 Nano kit (Agilent Technologies). Libraries were prepared using the TrueSeq RNA library prep kit v2 (Illumina) according to the manufacturer's instructions. Library qualities were assessed on a Bioanalyzer 2100 using a DNA 1000 kit (Agilent technologies). After quantification using the Qubit dsDNA High Sensitivity kit (Life technologies), libraries were pooled in equimolar quantities and sequenced on an Illumina NextSeq 550 (75 bp paired-end). Reads were mapped to mm10 genome with STAR v2.5.3a[62] allowing multimappers (–winAnchorMultimaNmax 100 –outFilterMultimapNmax 100 –outFilterMismatchNmax 3) and further processed with TEtranscrips[63] (–mode multi –stranded no) using GRCm38.95.gtf for transcript annotation and mm10_rmsk_TE.gtf for TE annotation. DESeq2[64] results were further analyzed in R v.3.6.0 environment. Significantly differentially regulated genes were defined as FDR-adjusted *p*-value < 0.05 and |FC| > 1.5. Data were filtered for genes that were significantly changed upon 6-h auxin treatment in E14; OsTIR1 cells. Differentially regulated transcripts are provided in Supplementary Data 2.

**Western blotting.** Cells were lysed in TOPEX buffer (50 mM Tris-HCl pH 7.5, 300 mM NaCl, 1 mM EDTA, 0.5% Triton X-100, 1% SDS, 1 mM DTT, 33.33 U/ml Benzonase with Halt Protease Inhibitor (Roche)). Proteins separated by SDS-PAGE using acrylamide gels (BioRad gel system), were transferred onto nitrocellulose membranes (Amersham[TM] Protran[TM] Premium 0.45 µm NC, GE Healthcare). The membranes were blocked in 5% skim milk (Sigma) in PBS-T (0.1% Tween-20 in PBS) and incubated with primary antibody of interest: anti-MPP8 (16796-1-AP, Proteintech, 1:500), anti-VINCULIN (SAB4200080, Sigma, 1:10000), anti-FLAG (F3165, Sigma, 1:5000), anti-ßACTIN (A2228, Sigma, 1:20000), anti-Cas9 (14697, Cell Signaling 1:1000), anti-OsTIR1 (PD048, MBL, 1:1000), anti-L1ORF1 (Ab216324, Abcam, 1:1000), anti-Tubulin (Ab176560, Abcam, 1:5000), anti-SETDB1 (Ab107225, Abcam, 1:500) (Supplementary Table 9). As secondary antibodies, either peroxidase-labeled anti-mouse (PI-2000, Vector Laboratories,

1:10000) and anti-rabbit IgG (PI-1000, Vector Laboratories, 1:10000) or IRDye 800CW Goat anti-Rabbit IgG (925-32211, LI-COR Bioscience, 1:10000) and IRDye 680RD Goat anti-Mouse IgG (926-68070, LI-COR Bioscience, 1:10000) were used. Proteins were detected by Super Signal West Pico chemiluminescent ECL substrate (Thermo Scientific) on a developer machine (Ferrania Imaging Technologies) or imaged using Image Studio Lite v.5.2.5 (Li-COR Biosciences). Uncropped western blots are provided in the Source data file.

**Immunoprecipitation followed by mass spectrometry.** Cells were lysed in IP300 buffer (50 mM Tris-HCl pH 7.5, 300 mM NaCl, 0.5% IGEPAL CA-630, 1 mM EDTA pH 8.0) supplemented with protease inhibitors (100 mM PMSF, 2 µg/ml leupeptin, 2 µg/ml aprotinin). Lysates were sonicated at 4 °C for five cycles setting (30 s ON/30 s OFF) followed by centrifugation at 20,000 × g for 30 min. Samples were pre-cleared with Protein G Sepharose beads incubated at 4 °C running end-over-end for 1 h. For Flag immunoprecipitation an aliquot of 5 mg protein in 5 ml IP300 buffer supplemented with protease inhibitors (IP300+) was incubated with FLAG M2 beads (Sigma) at 4 °C running end-over-end overnight. Samples were washed three times in IP300+ followed by three washes with TBS (50 mM Tris pH 7.5, 150 mM NaCl) supplemented with protease inhibitors, followed by three washes in ice-cold PBS. Beads were directly frozen on dry ice and kept at −80 °C until further use.

**Mass spectrometry.** Beads were resuspended in 20 µl lysis buffer (6 M Guanidinium Hydrochloride, 10 mM TCEP, 40 mM CAA, 50 mM HEPES pH 8.5), and sonicated 3 cycles 30 s on/off using a BioRupter Pico. Then, the samples were digested using a two-step digestion strategy; first, digestion buffer (10% Acetonitrile, 50 mM HEPES pH 8.5) was added with LysC (1:50 enzyme to protein ratio; MS grade, Wako) and incubated at 37 °C for 3 h, then with Trypsin (1:50 enzyme to protein ratio; MS grade, Promega) and incubated at 37 °C overnight. The next day, the reaction was quenched with 2% triformic acid (TFA; to a final of 1% TFA). C18 StageTips were packed in-house and activated with 4 washes: Methanol, Buffer B (80% Acetonitrile, 0.1% formic acid) and two washes of Buffer A' (3% Acetonitrile, 1% TFA). After filter capture through C18 tips the samples were washed twice with Buffer A (0.1% Formic Acid) and eluted with Buffer B (40% Acetonitrile, 0.1% Formic Acid). After concentration in Eppendorf Speedvac at 60 °C, the eluted fractions were reconstituted in Buffer A* (1% TFA, 2% Acetonitrile) for Mass Spectrometry analysis.

**Analysis of mass spectrometry data.** Label-free quantitation was used to derive protein abundances and resulting data were analyzed using Proteome Discoverer v.2.2 (Thermo Fisher Scientific). Statistical analysis was done in Perseus v.1.6.14.0[65]. In brief, protein abundances obtained from Proteome Discoverer were log2 transformed and results were filtered to have at least 2 valid values in each group. Missing values were imputed from the normal distribution using a width of 0.3 and down shift of 1.5. Volcano plots were used to reveal interactors using an FDR of 0.05 and $s_0$ of 2. Protein abundances and statistics are provided in Supplementary Data 3.

**Chromatin immunoprecipitation.** Cells were either double-crosslinked with disuccinimidyl glutarate (0.4 mM, Sigma-Aldrich) for 30 min followed by formaldehyde (1%, Sigma-Aldrich) for 10 min (for MPP8 (Proteintech, 16796-1-AP, 5 µl antibody to 300 µg chromatin), and FLAG (M2, Sigma, F3165) ChIP), or single-crosslinked using 1% formaldehyde for 10 min (for H3K9me3 (Abcam, Ab176916, 3 µl antibody to 50 µg chromatin), H3K4me3 (Cell signaling, 9751, 2 µl antibody to 50 µg chromatin) and H3K27ac (Active motive, 39685, 5 µl antibody to 50 µg chromatin) ChIP), neutralized by 0.125 M glycine and washed in PBS. Cells were collected in SDS buffer (50 mM Tris-HCl, pH 8, 100 mM NaCl, 5 mM EDTA, pH 8, 0.5% SDS, 0.02% NaN₃), centrifuged and resuspended in IP buffer (2 X volumes SDS buffer and 1 volume Triton dilution buffer (100 mM Tris-HCl pH 8, 100 mM NaCl, 5 mM EDTA pH 8, 5% Triton X-100, 0.02% NaN₃)) supplemented with protease inhibitors (2 µg/ml leupeptin, 2 µg/ml aprotinin). Samples were sonicated for 5 min (30 s ON/30 s OFF) in a Bioruptor (Diagenode). Chromatin was immunoprecipitated with the respective antibody, incubated overnight at 4 °C. The next day, protein G beads were added and incubated 4 °C running end-over-end for 3 h. Then, the beads were washed three times in low salt wash buffer (50 mM HEPES pH 7.5, 150 mM NaCl, 1% Triton X-100, 1 mM EDTA (pH 8.0), 0.1% NaDoc, 0.02% NaN₃), two times in high salt wash buffer (50 mM HEPES (pH 7.5,) 500 mM NaCl, 1% Triton X-100, 1 mM EDTA (pH 8.0), 0.1% NaDoc, 0.02% NaN₃) and once in IP buffer. Beads were incubated in elution buffer (100 mM NaHCO₃, 1% SDS) overnight at 65 °C shaking at 800 rpm for de-crosslinking. De-crosslinked DNA was purified using the Qiagen PCR Purification kit. Primers used for ChIP-qPCR are listed in Supplementary Table 10.

**Processing and analysis of ChIP-seq data.** Libraries were generated using the NEBNext ultra kit according to the manufacturer's specifications using Ampure XP beads (Beckman) for the size-selection steps. Libraries were sequenced on Illumina NextSeq 550 (75 bp paired-end). Sequencing data were demultiplexed and BCL files converted to FASTQ files using bcl2fastq2 v.2.20.0 (Illumina). Reads were first trimmed using Trim Galore v.0.4.5 (https://github.com/FelixKrueger/TrimGalore)

using –illumina and default parameters. Mapping was done with STAR v.2.5.3a/2.6. a[62] allowing multimappers (–winAnchorMultimaNmax 100 –outFilterMultimapNmax 100 –outFilterMismatchNmax 3) to mm10 assembly. Bam files were indexed using samtools v.1.10[66]. Peaks were called for using epic2 v.0.0.41[67] peak caller using the epic2-df function. Peaks were subsequently filtered to fulfill wild-type read number requirements (>= 100 && <=5000) and for peaks that are overrepresented in the wild-type condition (FC_WT > 1) yielding 55 high-confidence peaks. For visualization, bigWig coverage files normalized to sequencing depth *10000000 were generated with deepTools v.2.29.2[68] with the following configurations: bamCoverage –bam –ignoreDuplicates –scaleFactor –centerReads. BigWig were subsequently loaded into IGV v.2.8.0[69]. For average profiles, ngsplot v.2.63[70] was applied using default parameters. For the classification of activated LINE1 elements a threshold of 20 normalized counts was applied to include regions. Peak annotations were determined using Homer v.4.9[71] with default settings. For comparison, random peaks were sampled using bedtools v.2.27.1[59] shuffle with -noOverlapping option. Repeatmasker files used to annotate transposable elements in the mouse genome were downloaded from the M. Hammell lab (http://labshare.cshl.edu/shares/mhammelllab/www-data/TEtranscripts/TE_GTF/ mm10_rmsk_TE.gtf.gz)." Genomic coordinates and gene annotations of MPP8 binding sites are provided in Supplementary Data 4.

**Reporting summary**. Further information on research design is available in the Nature Research Reporting Summary linked to this article.

## Data availability
The mass spectrometry data have been deposited to the ProteomeXchange Consortium (http://proteomecentral.proteomexchange.org) via the PRIDE partner repository with the dataset identifier PXD019345. ChIP-seq and RNA-seq data have been submitted to the Gene Expression Omnibus (GEO) under accession code GSE150926. Source data are provided with this paper.

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

## Acknowledgements

The authors thank D. Schübeler for kindly providing *Dnmt* tKO mESCs, T. Jenuwein for *Suv39h* DKO mESCs and Y. Shinkai for *G9a/GLP* DKO and *Setdb1* cKO mESCs. We thank L.H. Blicher for technical assistance regarding the mass spectrometry experiments. We also thank all members of the Helin laboratory and R. Li for discussions and J.M. Brickman for critical reading of the manuscript. D.S. and A.R. were funded by the European Union's Horizon 2020 research and innovation program under the Marie Sklodowska-Curie grant agreements 749362 and 659171, respectively. The work in the Helin laboratory was supported by the Danish Cancer Society (grant no. R167-A10877), through a center grant from the NNF to the NNF Center for Stem Cell Biology (no. NNF17CC0027852), and through the Memorial Sloan Kettering Cancer Center Support Grant (no. NIH P30 CA008748).

## Author contributions

I.M. and K.H. conceived the study; I.M., A.S.M., T.T., and C.H. carried out experiments; E.M.S. conducted the mass spectrometry experiments and analysis; D.S., S.S., R.P.K., and I.M. performed the bioinformatics analysis; A.R. and J.H. provided intellectual support toward design and interpretation of the results; I.M. and K.H. wrote the original draft of the manuscript, followed by review and editing by all authors. K.H. supervised the study and acquired funding.

## Competing interests

K.H. is a consultant for Inthera Bioscience AG and a scientific advisor for Hannibal Innovation. All the other authors declare no competing interests.
