## [Peer Review File · Nature Communications]

REVIEWER COMMENTS

Reviewer #1 (Remarks to the Author):

Remarks to the authors:

Müller et al. performed a CRISPR/Cas9 screen using mouse ESCs (mESCs) and identified epigenetic factors implicated in the maintenance of mESCs under 2i/LIF condition. The authors then characterized one of the candidate factors MPP8 and its interacting proteins and demonstrate that MPP8 is critical for the survival of mESC in 2i/LIF, but not in serum. Using a proteomics approach, they identified MPP8 interacting proteins such as TASOR, PPHLN1 and SETDB1, each of which had been identified previously as members of the HUSH complex. The authors go on to apply biochemical approaches and genome-wide profiling to elucidate the roles of the N- and C-terminal regions of MPP8, and identify a key role for the C-terminal region (which lacks the chromodomain) in silencing of evolutionary young LINE1 elements. This is surprising, given that the N-terminal MPP8 chromodomain was previously proposed to be critical for its interaction with target sites via binding to H3K9me3 laid down by the H3K9 KMTase Setdb1. Based on the observations that both the repression of LINE1 and cell proliferation are restored by expressing C-terminal MPP8 in MPP8 deficient mESCs, the authors propose that it is this regulation of LINE1 activity that explains the critical role of MPP8 in maintenance of the ground state. While the manuscript is well-written and their finding provides mechanistic insights into the role of MPP8 in maintenance of ground state pluripotency and the roles of MPP8 domains and H3K9me3 in silencing of LINE1 elements in mESCs (related analyses have been published by other groups looking at other cell types in human and mouse), some of the claims made here should be further "fleshed out" before publication. Major and minor issues that should be addressed are outlined in detail below.

Major points:

1. While the authors claim that MPP8 depleted mESCs show failure of maintenance of "ground state", additional direct evidence should be provided, such as expression of key genes shown previously to be expressed exclusively in "ground state" ESCs. Such analyses (of their RNA-seq data) would directly demonstrate that maintenance of the "ground state" is affected in MPP8 deficient mESCs.
2. The authors demonstrate that MPP8 mutant mESCs cannot survive in 2i/LIF culture. However, it remains unclear whether MPP8 is required for promoting self-renewal or impeding differentiation, as the differentiation assay was done under serum/LIF. Could the authors clarify this issue by differentiating mESCs cultured under 2i/LIF?
3. It remains unclear how derepression of LINE1 elements in MPP8 deficient mESCs leads to cell death in 2i/LIF. In figure 2e, the authors observed defects in the cell cycle in the MPP8 mutant. Doesn't this suggest that cell cycle regulation may be a key function of MPP8, rather than LINE1 repression, in mESCs cultured in 2i/LIF?
4. Unfortunate and surprising given where the authors obtained the Suv39h1/2 DKO line that they do not cite Bulut-Karslioglu et al. (Mol Cell, 2014) here. In that paper Jenuwein and colleagues reported that young L1s and LTR elements are targeted for H3K9me3 by Setdb1 and Suv39h1/2, but only young L1s are upregulated in the absence of Suv39h1/2. They propose that Setdb1 seeds H3K9me3 at such L1s, while Suv39h1/2 play a role in spreading of this mark across these elements, which is apparently required for their silencing. No discussion of HUSH of course in this paper, but their observations would suggest that Suv39h1/2 are likely somehow involved in HUSH activity/spreading at these elements in particular, and perhaps MPP8 recruitment at these sites. Curiously, the authors show results for the Suv39h1/2 DKO in Figure 6 at specific genes, but not for L1 elements, which are

really the focus of the paper. Is MPP8 binding at their bona fide MPP8 targets, including those that include L1 elements, lost in the Suv39h1/2 DKO? ChIP qPCR at a minimum should be conducted to address this question.

5. It is surprising that there are only 55 MPP8 binding regions in the genome, given the number of TEs presumably targeted. A supplementary table listing the genomic coordinates and genic annotations for these binding sites if relevant should be included. While they attempted to show that evolutionary young LINE1 elements are enriched for MPP8 peaks in Figure S7a, it is not clear which of the young LINE1 families are truly enriched for these peaks. Does "all TEs" mean all TEs in the mouse genome? Does "Random peaks" mean random genomic regions having the same length as the MPP8 peaks were analyzed? There should be better way to present the significance of the results for each L1 family.

6. A profile of H3K9me3 should be added to Sup Fig 5C, to illustrate the relationship between these two marks at the bona fide MPP8 binding sites identified (shown later in S9). Do these regions correspond to the regions losing H3K9me3 in the Suv39h1/2 DKO as reported by Bulut-Karslioglu et al. (or in Setdb1 KO ESCs for that matter?)? Do the young L1 families reported to be upregulated in the Suv39h1/2 DKO by Bulut-Karslioglu et al. overlap with those reported to be upregulated here? Is the fold-change in expression in the Suv39h1/2 DKO vs MPP8 depleted lines similar?

7. Are the MPP8 peaks, including those with L1 elements, generally embedded within intragenic regions of active and/or inactive genes (which might also explain the few MPP8 bound regions identified)? This seems to be a feature of HUSH targets described in other studies, including a recent manuscript on BioRxiv (<https://www.biorxiv.org/content/10.1101/2020.03.09.974832v1.full>).

8. Related to the issues discussed above, while they mentioned in the legend of figure 4d-k that "Red dots: significantly changed LINE1 classes", their threshold for significance is unclear. Were the adjusted p-values for gene and TE transcripts calculated independently? If so, it might be confusing to plot dots for gene and TE transcripts in the same volcano plots (also hard to see some of the dots due to distinct dynamic ranges for genes and TEs). The authors could consider separating each of these volcano plots into two. While they observed profound (~8-fold) de-repression of L1orf2 by qPCR experiments, de-repression of LINE1 in MPP8 mutant mESCs detected by RNA-seq experiments seems modest- how many families up more than 2-fold? Might this be because most of the young LINE1 are efficiently silenced in MPP8 mutant mESCs, with only those LINE1 elements that are MPP8 bound de-repressed in the absence of MPP8? Agglomerated analysis of RNA-seq data may dilute the effect of MPP8 KO on LINE1 expression. Analysis of individual LINE1 elements could directly address this issue, but is of course complicated by the low mappability of the youngest elements.

9. The authors claim in the results section (page 15, lines 4-6) that they detected a reduction of H3K9me3 in MPP8 deficient cells at 48 hrs. However, the aggregation plots in Supplementary Fig. 9f (which is not referred to in the text) seem to show that not all of the MPP8 bound regions lose H3K9me3, at least not to the same extent. Are there regions that are bound by MPP8 where H3K9me3 is only modestly affected in the MPP8 deficient cells? If so, what is the difference in the regions that lose versus do not lose H3K9me3? How does this relate to the presence of L1 elements at these regions? Scatter plots showing ALL enriched regions would be more informative for a comprehensive/direct comparison of enrichment of H3K9me3 in control vs deficient cells than the aggregated plots and few screenshots of the same genic loci shown.

10. Is TASOR even expressed at the protein level in the absence of MPP8? In other words, is the lack of binding due to destabilization of TASOR here? A Western blotting showing this would be useful.

11. The authors show that both cell lines expressing N-terminally or C-terminally truncated MPP8

versions in MPP8 deficient mESCs failed to maintain H3K9me3 at MPP8 target sites and in turn conclude that, "since the N-terminally truncated MPP8 maintains the repression of LINE1 elements, these results demonstrate that H3K9me3 is not required for MPP8 to maintain the repression of these repeats". As they only see the H3K9me3 loss at 48 hrs, it is possible that during this time, the LINE1 elements in question GAIN DNA methylation, which then acts to suppress L1 expression! As the authors propose that it is likely that DNAm independently silences L1 elements in ESCs grown in serum/LIF, it is critical to rule out the possibility that silencing is maintained by the N-terminally truncated MPP8 as a consequence of gain of DNAm. Bisulfite Sanger-sequencing for "representative" regions would be useful to test if this is the case.

Similarly, it would be very informative for the authors to show what the status of Morc2 binding/enrichment is in their MPP8 deletion and rescue lines. Does the N-terminally truncated MPP8 maintain repression of LINE1 elements via Morc2 binding? As they clearly have gRNAs for Morc2 (Fig. 4e), KO of this HUSH-associated factor could be done in the MPP8 deletion/ N-terminally truncated MPP8 rescue background.

Minor points:

Abstract:

1. The authors state in the abstract that "the epigenetic regulation of....not fully understood". While this is true, as many studies have characterized the epigenetic regulation of ground state pluripotency in mESCs, the authors should at least cite these in the introduction (for example Marks et al., 2012 and van Mierlo et al., 2019).

2. "Epigenetic protein" and "epigenetic gene" are not really appropriate terms. These should be reworded.

Results:

1. Interesting that MPP8 was not among the genes identified by Li et al. (2018). Could the authors discuss this discrepancy?

2. Page 12 lines 23-24: "a strict correlation between the depression of LINE1 expression and loss of mESC self-renewal" is an overstatement. Strong evidence for this was not really presented.

3. Regarding genes upregulated in the MPP8 deficient cells, many studies have detected retrotransposons-initiated "chimeric" transcripts in cells depleted of factors involved in TE silencing, including SETDB1. Did the authors detect such "chimeric" transcripts among their upregulated genes? Do they overlap with previously identified transcripts?

4. The authors state on page 13 that "We observed differential regulation of 722 transcripts as early as six hours following MPP8 depletion, among these 396 arising from annotated genes and the remaining 326 from other annotated elements, such as repeats." These transcripts and their genomic locations/genes should be included in a supplementary table, divided according to their class.

5. Page 14 lines 15-16 The authors state: "Taken together, these results show that DNA methylation and MPP8 independently silence LINE1 expression in metastable mESCs, and that inactivation of both of these repressive mechanisms leads to growth arrest." Would be nice if the authors commented on the fact that while the DNMT TKO has less DNAm than WT ESCs cultured in 2i/LIF, the DNMT TKO cell growth phenotype in the absence of MPP8 is LESS severe than that observed for ESCs cultured in 2i/LIF in the absence of MPP8 (figure 1).

6. Would the authors care to speculate on the possibility that the N-terminus/chromodomain of MPP8 may be required for binding to SETDB1 due to methylation of ATF7IP, which was previously shown to be methylated (by G9a) and bound by MPP8 via the methylated residue, (Tsusaka et al., Epigenetics & Chromatin, 2018)?
7. The authors state in the Discussion on page 16 that "This observation also raises the exciting possibility that the HUSH complex displays an essential regulatory role in LINE1 control in other hypomethylated cell types, such as germ cells and cancer cells". Setdb1 at least has been deleted in both early germ cells (Liu et al, G&D, 2014) as well as growing oocytes (Eymery et al, Development, 2016), where DNAm levels are low, and little upregulation of L1 elements was observed in either. The authors should mention these findings.
8. In Figure 4e, each of the KOed factors should be aligned properly in the horizontal dimension, Atf7ip can simply be left blank in the MEF column.
9. H3K9me3 and DNAm tracks (the later published data in serum/LIF and 2i/LIF) should be added to Figure 4F. Really, it would be nice if these were added to each of the screen shots shown.
10. Label other significant dots in the Supplementary Fig 5a.
11. The culture conditions used for the expts presented in Figures 5 and 6 should be included in the figure/figure legend. This is critical to interpret the data as global DNAm levels differ in mESCs cultured in serum/LIF vs 2i/LIF.
12. For genome-browser screen shots with repeatmasker tracks included, where L1 elements are also labelled at the bottom, arrows should be added to indicate which of the repeats shown is actually the labelled/relevant L1 element. Adding genomic orientation of L1 elements would also help to compare the enrichment of H3K9me3 over the 5'-end/promoter or body of L1s in WT vs mutants.

Errata:

1. Figure 2e: IAA (500 M) should be corrected to IAA (500 μ M).
2. Figure 4f: "FLAG" is hard to see.
3. Figure 4h: "Refseq genes" should be removed as I believe this panel shows the enrichment of MPP8 around their peak regions.
4. Figure 5c: y-axis is missing.
5. Figure 6a,b: "Suv39h1/h1" should be corrected to "Suv39h1/h2".
6. Supplementary Fig. 9f is not referred to in the text.

Reviewer #2 (Remarks to the Author):

Muller et al (MS ID#: NCOMMS-20-20955)

In this study, the authors identified a protein named MPP8 as being essential for ground-state pluripotency. MPP8 is a component of the human silencing hub (HUSH) complex and has been suggested to repress LINE1 elements by recruiting the HUSH complex to H3K9me3-rich regions (Tchasovnikarova et al 2015 and Liu et al 2018). Interestingly and unexpectedly, the N-terminus deletion mutant of MPP8 can rescue the growth defect phenotype of the MPP8 depleted ES cells under the ground-stage culture condition (2i+LIF) and also suppress the HUSH-target LINE1 expression, but cannot accumulate on the HUSH-target loci. Furthermore, in the ES cells only expressing the N-terminus deletion mutant of MPP8, transcriptional repression of LINE1 elements is maintained without retaining H3K9me3 levels. Based on those findings, the authors propose that MPP8 silences LINE1 expression and protects the DNA-hypomethylated pluripotent ground state through its association with the HUSH core complex, however, stable chromatin binding and the maintenance of H3K9me3 is not essential for the MPP8-mediated LINE1 repression.

Overall, quality of presented data in this study is high and the manuscript is well written. The reviewer likes the epitope-tagged approach for biochemical and ChIP studies of endogenous molecules. The findings of the N-terminus and C-terminus deletion mutant of MPP8, MPP8112–858 and MPP81–729 are potentially interesting, for elucidating the molecular mechanism of the HUSH complex-mediated transcriptional silencing. Therefore, the reviewer is supportive for publication of this study in Nat Comm if the authors can address following critical issues.

Major comments,

1) MPP8 (WT and mutants) ChIP-seq and -qPCR analysis for the HUSH-target young LINE1 elements (derepressed copies by MPP8 depletion). This data is absolutely essential to show in this study even if both MPP8112–858 and MPP81–729 lost binding property to the HUSH-target LINE1 elements as shown for other MPP8-target loci such as Kcnq1 and Srrm2.

2) Fig. 4 MPP8 ChIP-seq analysis data and MPP8 WT 55 high confidence peaks.

The authors should describe more about distribution of the MPP8 WT 55 high confidence peaks. Genic, TSS, repeats,... Also, how many of them were mapped to the derepressed LINE1 elements? Supp Fig. 7 may describe this issue, but not enough explanation. Furthermore, to prove the ChIP-seq analysis data, the authors should perform MPP8 ChIP-qPCR analysis of the enriched LINE1 and show it, too.

3) Fig. 6c and Supp Fig. 9.

3)-1. Related to 1) and 2), H3K9me3 ChIP-qPCR analysis should be done to the derepressed different copies of young LINE1 elements such as L1Md_T and _Gf.

3)-2. Is this specific L1Md_T_dup5018 Lx9_dup 10388 analyzed in Fig. 6c really derepressed L1 in MPP8 depletion? If minor population of entire L1Md_T copies is derepressed by MPP8 depletion, it is difficult to deal with MPP8 targeting or H3K9me3 levels and MPP8-mediated LINE1 silencing issue because ChIP-seq reads of L1Md_T sequences seem to be multiply mapped regardless of transcriptional status. Should deal with this problem.

3)-3. Supp Fig. 9E and F (Aggregate plot comparing the average H3K9me3 ChIP-seq signal over all identified MPP8 peaks), these analysis should be done among only the MPP8-targeted LINE1 elements.

3)-4. Currently no clear mechanism how MPP8112–858 can suppress LINE1 expression without maintaining H3K9me3 level. At least, the authors should examine other epigenetic marks which have an impact on transcription such as acetylation, H3K4me2/3 in comparison with WT and MPP81–729.

4) Fig. 6A MPP8 ChIP-qPCR

Same with 1)~3), the MPP8-target LINE1 elements should be examined for the impact of SETDB1 or other H3K9 methyltransferases depletion. Also, "Data show one representative experiment (n = 1)" is not sufficient to make their conclusion. Provide additional evidences.

5) How about LINE1 depression in the MPP8 depleted ES cells under the culture condition with single

addition of GSK inhibitor which did not induce growth defect by MPP8 depletion? This may provide additional insight about the role of MPP8 for ground-state pluripotency and LINE1 regulation.

6) MPP81-729

RNA-seq analysis at 48 hours post auxin treatment, the phenotype of MPP8 depleted cells and MPP8 depleted cells complemented with MPP81-729 is quite different. Later cells induced 10x more up- or down-regulated transcripts (Non rescued: 314, MPP81-729: 4481). Furthermore, MPP81-729 showed more strong impact than simple MPP8 depletion, or MPP81-522 and MPP81-188 rescued cells (Fig. 2c and Fig. 3d), suggesting that MPP81-729 is not a simple recessive mutant, but a dominant negative one. The authors should discuss this in the text.

Minor comments,

1) In introduction, "Moreover, MPP8 has been shown to interact with multiple epigenetic silencing proteins, including the H3K9 mono- and di-methyltransferase proteins GLP/G9a11,12, DNA methyltransferase DNMT3A11,12 and histone deacetylase SIRT113."

ATF7IP should be included. ATF7IP methylated by G9a/GLP is also shown to bind MPP8 (Tsusaka et al 2018).

Point-to-point response to the Referees

We thank both reviewers for their overall positive criticism and constructive input. We have now carefully revised our manuscript, and included new figure panels (Fig. 2h-j, Supplementary Fig. 5b,d,e,h,i, Supplementary Fig. 7f,g, Supplementary Fig. 8b, Supplementary Fig. 10b,c,e,f) and tables (Supplementary Data 3,4), as well as the revised several other figures panels. Moreover, we have thoroughly integrated suggestions for improvements in the introduction, results and discussion sections. Please find our point-by-point responses to all major and minor comments below.

Reviewer #1 (Remarks to the Author):

Remarks to the authors:

Müller et al. performed a CRISPR/Cas9 screen using mouse ESCs (mESCs) and identified epigenetic factors implicated in the maintenance of mESCs under 2i/LIF condition. The authors then characterized one of the candidate factors MPP8 and its interacting proteins and demonstrate that MPP8 is critical for the survival of mESC in 2i/LIF, but not in serum. Using a proteomics approach, they identified MPP8 interacting proteins such as TASOR, PPHLN1 and SETDB1, each of which had been identified previously as members of the HUSH complex. The authors go on to apply biochemical approaches and genome-wide profiling to elucidate the roles of the N- and C-terminal regions of MPP8, and identify a key role for the C-terminal region (which lacks the chromodomain) in silencing of evolutionary young LINE1 elements. This is surprising, given that the N-terminal MPP8 chromodomain was previously proposed to be critical for its interaction with target sites via binding to H3K9me3 laid down by the H3K9 KMTase Setdb1. Based on the observations that both the repression of LINE1 and cell proliferation are restored by expressing C-terminal MPP8 in MPP8 deficient mESCs, the authors propose that it is this regulation of LINE1 activity that explains the critical role of MPP8 in maintenance of the ground state. While the manuscript is well-written and their finding provides mechanistic insights into the role of MPP8 in maintenance of ground state pluripotency and the roles of MPP8 domains and H3K9me3 in silencing of LINE1 elements in mESCs (related analyses have been published by other groups looking at other cell types in human and mouse), some of the claims made here should be further "fleshed out" before publication. Major and minor issues that should be addressed are outlined in detail below.

Major points:

1. While the authors claim that MPP8 depleted mESCs show failure of maintenance of "ground state", additional direct evidence should be provided, such as expression of key genes shown previously to be expressed exclusively in "ground state" ESCs. Such analyses (of their RNA-seq data) would directly demonstrate that maintenance of the "ground state" is affected in MPP8 deficient mESCs.

We do not think that MPP8 works as a direct transcriptional master regulator of ground state genes but rather that loss of the ground state is mediated through indirect effects that are secondary to LINE1 expression, as G1 arrest (Fig. 2e) and later changes of ground-state pluripotency genes which are incompatible with cell survival in 2i/LIF culture conditions (Fig. 2c).

However, as suggested, we re-examined the RNA-seq data at 6 hours after MPP8 removal for genes classified as naïve pluripotency genes by Kalkan et al., 2019, and in agreement with our model, we did not see destabilization of the ground state transcriptional network at this timepoint. Importantly, however, when inducing exit from 2i culture, we indeed observed destabilization of the ground state pluripotent network upon loss of MPP8 (see question 2, Fig. 2i,j).

2. The authors demonstrate that MPP8 mutant mESCs cannot survive in 2i/LIF culture. However, it remains unclear whether MPP8 is required for promoting self-renewal or impeding differentiation, as the differentiation assay was done under serum/LIF. Could the authors clarify this issue by differentiating mESCs cultured under 2i/LIF?

To address this question, we have designed an experiment similar to the setup in Kalkan et al., 2017, Development: mESCs were cultured in 2i only, since the addition of LIF to the media delays the onset of differentiation, and were then released into N2B27 media without 2i in presence or absence of MPP8 (Fig. 2h). We found that MPP8 protects naïve pluripotency by contributing to the transcriptional control of pluripotency markers while moderately influencing the expression kinetics of differentiation makers (Fig. 2i,j).

We have included the following paragraph and the figure panels below in the manuscript: “However, when monitoring the early phase of mESC transition after withdrawal from 2i in serum-free N2B27 medium (Fig. 2h), MPP8 loss led to a fast destabilization of the naïve gene expression program upon (Fig. 2i), while no accelerated upregulation of differentiation markers was observed (Fig. 2j). These results suggest that the primary effect of MPP8 is to promote self-renewal of naïve mESCs.”

3. It remains unclear how derepression of LINE1 elements in MPP8 deficient mESCs leads to cell death in 2i/LIF. In figure 2e, the authors observed defects in the cell cycle in the MPP8 mutant. Doesn't this suggest that cell cycle regulation may be a key function of MPP8, rather than LINE1 repression, in mESCs cultured in 2i/LIF?

The reviewer is correct in noting that MPP8 depletion leads to cell cycle effects (Fig 2e): our data showed a trend towards an accumulation of cells in G1 phase and a depletion of cells in S phase already 6 hours after MPP8 depletion and a significant G1 arrest 48 hours post MPP8 depletion. These changes are also reflected in the gene expression profiles. GO analysis of the list of differentially regulated transcripts at 6 hours post MPP8 depletion using the GSEA software (<https://www.gsea-msigdb.org/>) showed that upregulated genes were significantly enriched for cell cycle-related genes (Peer Review Figure 1).

Peer Review Figure 1: Top 10 enriched GO biological processes for upregulated genes 6 hours post MPP8 depletion

However, our results do not support that MPP8 directly regulates cell cycle-related genes, because we have not detected MPP8 binding to these genes. In contrast, we find MPP8 is enriched on evolutionary young LINE1 elements and that LINE1 elements are upregulated few hours after MPP8 depletion. Thus, our data confirms that evolutionary young LINE1 elements are direct targets of MPP8 in mESCs. Based on this we speculate that deregulation of LINE1 expression leads to effects on the cell cycle and induces cell death as secondary response. This

hypothesis is indeed in agreement with a recent report that showed G1 arrest as consequence of ectopic LINE1 overexpression in RPE cells (Areljan et al., 2020).

We have summarized this in the discussion section:

“Moreover, we show that MPP8 is required for the self-renewal of ground-state mESC and suggest that the loss of self-renewal upon loss of MPP8 is caused by an increase in LINE1 expression. This hypothesis is in agreement with recent reports demonstrating that ectopic LINE1 overexpression leads to G1 arrest in RPE cells⁴⁷.”

4. Unfortunate and surprising given where the authors obtained the Suv39h1/2 DKO line that they do not cite Bulut-Karslioglu et al. (Mol Cell, 2014) here.

We thank the referee for noticing this shortcoming. We have now added the citation.

In that paper Jenuwein and colleagues reported that young L1s and LTR elements are targeted for H3K9me3 by Setdb1 and Suv39h1/2, but only young L1s are upregulated in the absence of Suv39h1/2. They propose that Setdb1 seeds H3K9me3 at such L1s, while Suv39h1/2 play a role in spreading of this mark across these elements, which is apparently required for their silencing. No discussion of HUSH of course in this paper, but their observations would suggest that Suv39h1/2 are likely somehow involved in HUSH activity/spreading at these elements in particular, and perhaps MPP8 recruitment at these sites. Curiously, the authors show results for the Suv39h1/2 DKO in Figure 6 at specific genes, but not for L1 elements, which are really the focus of the paper.

Is MPP8 binding at their bona fide MPP8 targets, including those that include L1 elements, lost in the Suv39h1/2 DKO? CHIP qPCR at a minimum should be conducted to address this question.

We thank the reviewer for this important comment, and have now extended Fig. 6a and 6b showing two peaks that overlap LINE1 elements as well two non-LINE1 MPP8 binding sites and two negative control loci not bound by MPP8. As for H3K9me3, we did not observe an impact of *Suv39h1/h2* dKO on H3K9me3 deposition on any of the investigated MPP8 binding sites, while we see a reduction of H3K9me3 after *Setdb1* KO, hence suggesting that SETDB1 is the major H3K9me3 methyltransferase at MPP8-bound regions. Similarly, while we observed a marked impact of *Setdb1* KO on MPP8 recruitment, consistent with a published role of SETDB1 upstream of MPP8 binding. In contrast, we did not detect a decrease in MPP8 binding in *Suv39h1/h2* dKO cells.

We now further describe and discuss this point in the result section:

“While Suv39h1/h2-mediated H3K9me3 has been shown to contribute to silencing of evolutionary young LINE1 elements in serum/LIF-grown mESCs³⁹, we did not find evidence for a role of Suv39h1/h2 at MPP8-bound regions, as neither H3K9me3 levels (Fig. 6a) nor MPP8 recruitment (Fig. 6b) were altered in *Suv39h1/h2* dKOs at the investigated target loci. Consistent with a described role for SETDB1 in MPP8 chromatin binding^{16,40}, its deletion led to loss of MPP8 recruitment to all investigated target loci (**Fig. 6a** and **Supplementary Fig. 9a**) concomitant with H3K9me3 reduction (**Fig. 6b**). Therefore, we conclude that SETDB1 is both the main H3K9me3 methyltransferase responsible for H3K9me3 deposition as well as MPP8 recruitment at MPP8-bound regions in ground-state mESCs.”

5. It is surprising that there are only 55 MPP8 binding regions in the genome, given the number of TEs presumably targeted.

The low number of bound TEs could potentially be explained by the low mapping sensitivity to these highly repetitive regions which are notoriously difficult to map. Also, we show that the nature of HUSH complex binding to chromatin is potentially more transient and detectable binding is not a prerequisite for silencing. Hence, there might be more MPP8 binding sites that are not retrievable by current ChIP methods. Moreover, the stringent criteria we applied for the identification of peaks (e.g. fold change over knockout >2) and potential incomplete removal of MPP8 binding after IAA treatment as suggested by our qPCR data (Supplementary Fig. 5b), might also have contributed to filtering of a fraction of true positive MPP8 binding sites. As we are aware of these technical challenges, we do not claim completeness for the identification of MPP8 binding sites in mESCs but refer to them as 'high confidence MPP8 peaks'.

A supplementary table listing the genomic coordinates and genic annotations for these binding sites if relevant should be included.

As requested, we have added a table containing the genomic coordinates and genic annotations of MPP8 binding sites in Supplementary Data 3.

While they attempted to show that evolutionary young LINE1 elements are enriched for MPP8 peaks in Figure S7a, it is not clear which of the young LINE1 families are truly enriched for these peaks. Does "all TEs" mean all TEs in the mouse genome? Does "Random peaks" mean random genomic regions having the same length as the MPP8 peaks were analyzed? There should be a better way to present the significance of the results for each L1 family.

We have now chosen a different way to present the results for the top 10 enriched LINE1 classes overlapping MPP8 peaks which in our opinion better highlights the enrichment of evolutionary young LINE1 classes, as L1Md_A and L1Md_T compared to their expected fraction in the mouse genome or random sampled peaks. The random peaks were sampled using bedtools (version 2.27.1) shuffle with -noOverlapping option. This resulted in 55 random peaks with the same mean length as MPP8 peaks.

Moreover, we have added a more detailed description of the bioinformatics approach to the method section:

"For comparison, random peaks were sampled using bedtools (Version 2.27.1) shuffle with -noOverlapping option. Repeatmasker files used to annotate transposable elements in the mouse genome were downloaded from the M. Hammell lab (http://labshare.cshl.edu/shares/mhammelllab/www-data/TEtranscripts/TE_GTF/mm10_rmsk_TE.gtf.gz)."

6. A profile of H3K9me3 should be added to Sup Fig 5C, to illustrate the relationship between these two marks at the bona fide MPP8 binding sites identified (shown later in S9).

The average H3K9me3 ChIP signal over MPP8 binding sites has now been added to the same figure (Supplementary Fig. 5d).

Do these regions correspond to the regions losing H3K9me3 in the Suv39h1/2 DKO as reported by Bulut-Karslioglu et al. (or in Setdb1 KO ESCs for that matter)?

To address this question, we re-analyzed the data published by Bulut-Karslioglu et al., 2014 using our analysis pipeline and identified 51677 Suv39h1/h2-dependent H3K9me3 peaks. These showed a great difference between average binding profiles of H3K9me3 in wildtype and Suv39h1/h2 KO cells, as expected (*Peer Review Figure 2*). Over MPP8 binding sites, the H3K9me3 signal was reduced but not completely lost in Suv39h1/h2 dKO, suggesting a potential contribution of Suv39h1/h2 to maintenance of H3K9me3 at MPP8 binding loci (*Peer*

Review Figure 2b). This effect, however, is less pronounced than the observed loss of H3K9me3 in *Setdb1* KO cells (Fig. 6b), indicating that SETDB1 is the major H3K9me3 methyltransferase at MPP8 binding sites. It is also to be kept in mind that while for both MPP8 and *Setdb1* knockout is conditionally induced on protein or DNA level, respectively, *Suv39h1/h2* dKO cells are kept as constitutive KO cells, and the cells may therefore have adapted to this condition. Moreover, Bulut-Karslioglu et al. maintained the mESCs in serum/LIF and the distribution of H3K9me3 the role of *Suv39h1/h2* in catalyzing H3K9me3 may be different in 2i/LIF. Future work addressing the impact of *Suv39h1/h2* removal on the kinetics of H3K9me3 and MPP8 recruitment to MPP8 binding sites using an inducible *Suv39h1/h2* dKO cell line maintained in 2i/LIF will be of great interest for the field.

Peer Review Figure 2: Comparison of H3K9me3 ChIPSeq in *Suv39h* dKO (Bulut-Karslioglu et al., 2014) and *MPP8^{mAID};OsTIR1* cells 48 hours of IAA treatment .a-b) Aggregate plot comparing the average H3K9me3 ChIP signal over (a) *Suv39h*-dependent H3K9me3 peaks ($n=51677$) and (b) high confidence MPP8 peaks ($n=55$) in wt E14 cells or *Suv39h1/h2* dKO cells. Input signal serves as control. Data is taken from Bulut-Karslioglu et al., 2014.

Do the young L1 families reported to be upregulated in the *Suv39h1/2* DKO by Bulut-Karslioglu et al. overlap with those reported to be upregulated here? Is the fold-change in expression in the *Suv39h1/2* DKO vs MPP8 depleted lines similar?

Bulut-Karslioglu et al. reported seven LINE1 classes to be significantly upregulated in the *Suv39h1/2* dKO cells, for which only one class, the evolutionary young L1Md_A class, showed an upregulation of greater than 2-fold (Peer Review Table 1).

Suv39h1/h2 dKO (Serum/LIF)		
LINE class	Fold change	padj
L1Md_A	3.10	9.53E-245
Lx3_Mus	1.27	3.06E-07
Lx2B	1.24	2.91E-09
L1_Mm	1.17	1.91E-04
L1Md_T	1.16	1.78E-03
L1Md_F	1.15	8.15E-04
HAL1	1.12	7.85E-03

Peer Review Table 1: LINE1 classes defined as significantly upregulated in Bulut-Karslioglu et al., 2014. In *Suv39h1/h2* dKO mESCs compared to wt mESCs. L1md_A is highlighted in bold as the only class with fold upregulation > 2.

Comparing significantly upregulated LINE1 classes in *Suv39h1/h2* dKOs to those significantly upregulated at 6 hours after MPP8 depletion, three LINE1 classes overlap, among them the evolutionary young L1Md_A and L1Md_T (Peer Review Figure 3). The majority of LINE1 classes displays higher up-regulation in MPP8-depleted mESCs compared to *Suv39h1/h2* dKO ESCs: While only one LINE1 class, the evolutionary young L1Md_A class, reached over 2-fold upregulation in *Suv39h1/h2* dKO ESCs, four LINE1 classes were more than 2-fold upregulated after 6 hours of MPP8 removal (Peer Review Figure 3a,b). After 48 hours of MPP8 depletion, three significantly upregulated LINE1 classes overlap with those observed in *Suv39h1/h2* dKOs (Peer Review Figure 3c,d). Again, apart from L1Md_A, upregulation levels observed in MPP8-depleted cells are more pronounced than those observed in *Suv39h1/h2* dKOs (Peer Review Figure 2d).

Peer Review Figure 3: Comparison of significantly upregulated LINE1 classes in *Suv39h1/h2* dKO mESCs (Bulut-Karslioglu et al., 2014) and *MPP8^{mAID};OsTIR1* mESCs in response to 6 hours or 48 hours of IAA treatment. a) Overlap of significantly upregulated LINE1 classes in *Suv39h1/h2* dKO mESCs and *MPP8^{mAID};OsTIR1* mESCs upon 6 hours of IAA treatment (500 μ M) as detected by polyA-selected RNAseq. b) Upregulation of *Suv39h1/h2*-regulated LINE1 classes in MPP8-depleted (6 hours IAA) mESCs. Fold changes of significantly upregulated LINE classes in *Suv39h1/h2* dKO ESCs were plotted against the fold changes of the same LINE classes observed after MPP8 depletion for 6 hours (only LINE classes are plotted for which fold changes were determined in both data sets). c) Overlap of significantly upregulated LINE1 classes in *Suv39h1/h2* dKO mESCs and *MPP8^{mAID};OsTIR1* mESCs upon 48 hours of IAA treatment (500 μ M) as detected by polyA-selected RNAseq. d) Upregulation of *Suv39h*-regulated LINE1 classes in MPP8-depleted (48 hours IAA) ESCs. Fold changes of significantly upregulated LINE classes in *Suv39h* dKO ESCs were plotted against the fold changes of the same LINE classes observed after MPP8 depletion for 48 hours (only LINE classes are plotted for which fold changes were determined in both data sets).

7. Are the MPP8 peaks, including those with L1 elements, generally embedded within intragenic regions of active and/or inactive genes (which might also explain the few MPP8 bound regions identified)? This seems to be a feature of HUSH targets described in other studies, including a recent manuscript on BioRxiv

(<https://www.biorxiv.org/content/10.1101/2020.03.09.974832v1.full>).

We have now added a figure showing the genomic distribution of MPP8-bound peaks (Supplementary Fig. 5e). We find the majority of peaks located in intergenic regions (60%) followed by introns (22%).

8. Related to the issues discussed above, while they mentioned in the legend of figure 4d-k that “Red dots: significantly changed LINE1 classes”, their threshold for significance is unclear. Were the adjusted p-values for gene and TE transcripts calculated independently?

To quantify both gene and TE transcript abundances, we have used the Tetranscripts tool from the Hammell lab (Jin et al., 2015 Bioinformatics). As described in the workflow diagram, counts for genes and TEs were analyzed together by DEseq2 and hence the same thresholds were used (<http://hammellab.labsites.cshl.edu/software/#Tetranscripts>).

If so, it might be confusing to plot dots for gene and TE transcripts in the same volcano plots (also hard to see some of the dots due to distinct dynamic ranges for genes and TEs). The authors could consider separating each of these volcano plots into two.

As described above, p-values have been calculated simultaneously for genes and TE transcripts which is why we prefer to show them in the same plot. Additionally, to give a better overview of upregulated LINE1 classes after 6 hours and 48 hours of MPP8 removal, we have added Supplementary Fig. 7f.

While they observed profound (~8-fold) de-repression of L1orf2 by qPCR experiments, de-repression of LINE1 in MPP8 mutant mESCs detected by RNA-seq experiments seems modest-how many families up more than 2-fold?

We have now added an overview table summarizing upregulated LINE1 classes at 6 hours and 48 hours post MPP8 depletion (Supplementary Fig. 7f). At 6 hours post MPP8 depletion, nine LINE1 classes were significantly ($\text{padj} < 0.05$) upregulated, four of which showed upregulation greater than 2-fold. At 48 hours post MPP8 depletion, four LINE1 classes were significantly ($\text{padj} < 0.05$) upregulated, all of which showed upregulation levels greater than 2-fold (Fig 5d). As L1orf2 transcripts assessed by qPCR in Fig. 5a are derived from different classes of evolutionary young LINE1 families (as e.g. L1Md_Gf, L1Md_T, L1Md_A), we consider the observed level of upregulation for L1orf2 transcripts by qPCR well reflected in the combined upregulation of single young LINE1 classes as observed by RNA-seq.

Might this be because most of the young LINE1 are efficiently silenced in MPP8 mutant mESCs, with only those LINE1 elements that are MPP8 bound de-repressed in the absence of MPP8?

Given the low number of (detectable) MPP8 binding sites versus the large number of LINE1 elements encoded in the mouse genome, we indeed believe that the majority of LINE1 elements remains efficiently silenced in MPP8 mutant cells after 6 hours of MPP8 removal by mechanisms that are independent of MPP8. In agreement with this hypothesis, a zinc finger protein has been suggested as recruitment mechanism upstream of SETDB1 for silencing of unintegrated retroviral DNA (Zhu et al. 2018, Nature). Similarly, specific zinc finger proteins might be responsible to recruit SETDB1 and in turn MPP8 to its binding sites in mESCs,

potentially explaining the specific targeting of a subset of LINE1 elements. The upregulation of LINE1 classes observed after 48 hours of MPP8 removal probably presents a mixture of direct effects caused by loss of MPP8 and indirect effects linked to the observed phenotype which is prominent at 48 hours of MPP8 removal.

Agglomerated analysis of RNA-seq data may dilute the effect of MPP8 KO on LINE1 expression. Analysis of individual LINE1 elements could directly address this issue, but is of course complicated by the low mappability of the youngest elements.

As the referee states, mapping of repetitive elements to unique loci is notoriously challenging, especially of the evolutionary youngest classes, which are most conserved in their sequence (Furano et al. 2000, NAR.) As we observed upregulation of specifically the youngest LINE1 elements using agglomerated RNA-seq analysis (Fig. 5d-k), we were technically limited to class-based mapping instead of mapping of the reads to individual LINE1 loci. In future studies, we are planning on testing the feasibility of using currently emerging technological advancements that increase the mappability of repetitive elements, as for instance the use of nanopore sequencing.

9. The authors claim in the results section (page 15, lines 4-6) that they detected a reduction of H3K9me3 in MPP8 deficient cells at 48 hrs. However, the aggregation plots in Supplementary Fig. 9f (which is not referred to in the text)

We have now referred to Fig. 9f in the text

seem to show that not all of the MPP8 bound regions lose H3K9me3, at least not to the same extent. Are there regions that are bound by MPP8 where H3K9me3 is only modestly affected in the MPP8 deficient cells? If so, what is the difference in the regions that lose versus do not lose H3K9me3? How does this relate to the presence of L1 elements at these regions? Scatter plots showing ALL enriched regions would be more informative for a comprehensive/direct comparison of enrichment of H3K9me3 in control vs deficient cells than the aggregated plots and few screenshots of the same genic loci shown.

As the reviewer notes, when visually comparing different MPP8 bound regions, we indeed observed that the extend of reduction differs from region to region as exemplified when comparing Fig. 6d with Supplementary Fig. 9d. To address if the presence of LINE1 elements at these regions correlates with the level of H3K9me3 reduction, we have added aggregate plots showing average levels of H3K9me3 over MPP8-targeted LINE1 regions and MPP8-targeted regions that do not overlap upregulated LINE1 classes (Supplementary Fig. 10c,d,e,f). We prefer showing aggregated plots over scatter plots due to the low number of MPP8-bound regions. The comparison revealed similar average H3K9me3 reduction levels independent of the presence or absence of LINE1 elements at MPP8 target sites. Hence, we conclude that the presence of LINE1 elements is not a distinctive determinant for H3K9me3 kinetics at MPP8 binding sites upon loss of MPP8. While we do not know what determines H3K9me3 kinetics at these regions, we speculate that both direct and indirect effects may contribute at this later timepoint after MPP8 removal.

10. Is TASOR even expressed at the protein level in the absence of MPP8? In other words, is the lack of binding due to destabilization of TASOR here? A Western blotting showing this would be useful.

We indeed observed a decrease in TASOR protein levels in the absence of endogenous MPP8. Furthermore, we show that the stability of TASOR is dependent on the C-terminal part of MPP8

that is required for the binding of MPP8 to TASOR. We have added the corresponding western blot in Supplementary Fig. 5b, and the following text in the manuscript:

“In fact, we observed a decrease in TASOR protein levels upon degradation of MPP8, which is in agreement with previous observations in HeLa cells¹⁶. Moreover, we showed that the stability of TASOR is dependent on the C-terminal part of MPP8, which is required for the binding to TASOR (**Supplementary Fig. 5b**).”

11. The authors show that both cell lines expressing N-terminally or C-terminally truncated MPP8 versions in MPP8 deficient mESCs failed to maintain H3K9me3 at MPP8 target sites and in turn conclude that, “since the N-terminally truncated MPP8 maintains the repression of LINE1 elements, these results demonstrate that H3K9me3 is not required for MPP8 to maintain the repression of these repeats”. As they only see the H3K9me3 loss at 48 hrs, it is possible that during this time, the LINE1 elements in question GAIN DNA methylation, which then acts to suppress L1 expression! As the authors propose that it is likely that DNAm independently silences L1 elements in ESCs grown in serum/LIF, it is critical to rule out the possibility that silencing is maintained by the N-terminally truncated MPP8 as a consequence of gain of DNAm. Bisulfite Sanger-sequencing for “representative” regions would be useful to test if this is the case.

The reviewer is correct in suggesting that DNA methylation may substitute for the loss of H3K9me3 on the LINE1 elements, however, this does not change our conclusion that H3K9me3 is not required to maintain the repression of these repeats. Nevertheless, we would have liked to perform the experiments suggested by the reviewer, however, due to the highly repeated DNA sequence, it is not technically possible to design specific primers for bisulfite Sanger-sequencing on the evolutionary youngest LINE1 elements targeted by MPP8, for which we observed highest upregulation level on qPCR. Moreover, because we do not observe binding of MPP8 to all LINE1 elements, but only to a specific subset, as evidenced by the low number of MPP8 peaks as well as the low ChIP-qPCR enrichment of MPP8 on specific LINE1 classes (*Peer Review Fig. 4*), bisulfite Sanger-sequencing using primers that amplify LINE1 classes will not provide the sensitivity to capture DNA methylation changes on MPP8-bound LINE1 elements.

Peer Review Figure 4: Enrichment of MPP8 on bulk L1 promoter regions.

Similarly, it would be very informative for the authors to show what the status of Morc2 binding/enrichment is in their MPP8 deletion and rescue lines. Does the N-terminally truncated MPP8 maintain repression of LINE1 elements via Morc2 binding?

To analyze the potential overlap between MPP8 and MORC2A binding, we have re-analyzed published MORC2A ChIPSeq data generated in mESCs cultured in Serum/LIF (Fukada et al., 2018) using our analysis pipeline. The analysis revealed a significant overlap between MORC2A and MPP8 peaks in mESCs, as previously shown in HeLa cells (Tchasovnikarova et al., 2017), supporting a potential role of MORC2A in HUSH-mediated silencing at these loci in mESCs (Fig. 4f, Supplementary Fig. 5g-h). These observations together with upregulation of

evolutionary young LINE elements upon sgRNA-mediated targeting of *Morc2a* in mESCs as previously shown (Fukada et al., 2018) indicate a potential requirement of MPP8 for MORC2A binding. To test the hypothesis whether MORC2A is recruited to MPP8 target loci in presence of MPP8¹¹²⁻⁸⁵⁸ we tried overexpressing HA-MORC2A into the Mpp8mAID; OsTIR1 genetic background given the absence of a suitable antibody for chromatin immunoprecipitation of endogenous mouse MORC2A. However, we were not able to express sufficient levels of MORC2A to assess chromatin binding. Although we believe it could be interesting to include results on the MORC2A-MPP8 binding in this manuscript, we find that elucidating the precise silencing mechanism is beyond the scope of the current study.

As they clearly have gRNAs for *Morc2* (Fig. 4e), KO of this HUSH-associated factor could be done in the MPP8 deletion/ N-terminally truncated MPP8 rescue background.

As we show in Fig. 4e, gRNA mediated targeting of *Morc2a* impairs mESC proliferation under 2i/LIF culture conditions and hence generation of stable *Morc2a* KO cells is likely not possible independent of the genetic status of *Mphosph8*.

Minor points:

Abstract:

1. The authors state in the abstract that “the epigenetic regulation of....not fully understood”. While this is true, as many studies have characterized the epigenetic regulation of ground state pluripotency in mESCs, the authors should at least cite these in the introduction (for example Marks et al., 2012 and van Mierlo et al., 2019).

We thank the referee for pointing out this shortcoming. We have now cited the suggested literature in the introduction.

“However, genome-wide studies of epigenetic modifications of chromatin⁸ and proteomic profiling of chromatin-associated complexes and histone modifications⁹ have revealed distinct features of the ground-state epigenome, indicating a unique contribution of epigenetics for ground-state pluripotency.”

2. “Epigenetic protein” and “epigenetic gene” are not really appropriate terms. These should be reworded.

We have now reworded the terms with ‘potential regulators of epigenetic processes’, ‘proteins involved in epigenetic processes’, ‘proteins with a potential role in regulating epigenetic features’ and ‘genes encoding proteins with potential epigenetic function’

Results:

1. Interesting that MPP8 was not among the genes identified by Li et al. (2018). Could the authors discuss this discrepancy?

One potential explanation for this, is that we generated a more sensitive library by using a domain-targeting approach which was previously shown to enhance negative selection screens (Shi et al. 2015, Nat. Biotechnol), whereas Li et al. chose to target mostly 5'exons. Moreover, we used an epigenetics-focused, instead of a genome-wide library allowing us to include a higher number of sgRNAs (on average 10 sgRNAs) per gene compared to the genome-wide approach used in Li et al. Specifically looking at the *Mphosph8* locus, we included 10 sgRNAs in the library that are placed within either of the two annotated domains (6 within the chromodomain and 4 within the ankyrin repeat domain) (Fig. 3a), while Li et al. included 5 sgRNAs targeting either of the first 3 annotated exons, hence placing the sgRNAs either upstream (1 sgRNA), within (2 sgRNAs) or

immediately downstream (2 sgRNAs) of the chromodomain. As shown in Fig. 3a, we observed strongest effects on mESC proliferation for sgRNAs placed in the ankyrin repeat domain. This is most likely due to their immediate upstream proximity to the critical C-terminal region which we find to interact with TASOR and to be required for self-renewal of ground-state mESCs. The sgRNAs placed within the chromodomain, on contrary, show much lower depletion levels in the screen (< 5 fold) and hence would not have identified MPP8 as hit using our selection criteria.

2. Page 12 lines 23-24: “a strict correlation between the depression of LINE1 expression and loss of mESC self-renewal” is an overstatement. Strong evidence for this was not really presented.

We have now rephrased the sentence:

“These results suggest a potential correlation between de-repression of LINE1 elements and loss of mESC self-renewal.”

3. Regarding genes upregulated in the MPP8 deficient cells, many studies have detected retrotransposons-initiated “chimeric” transcripts in cells depleted of factors involved in TE silencing, including SETDB1. Did the authors detect such “chimeric” transcripts among their upregulated genes? Do they overlap with previously identified transcripts?

We thank the reviewer for this comment. We have now investigated the presence of gene-TE chimeric transcripts in MPP8 deficient cells. We observed increased numbers of gene-TE chimeric transcripts upon 6 hours of MPP8 depletion, most of which could be rescued by re-expression of MPP8^{wt} (Peer Review Fig. 5a). When looking at all chimeric transcripts enriched upon MPP8 removal, a broad range of repeat elements across all TE families could be identified (Peer Review Fig. 5b). Inspection at coordinate level revealed the presence of several interesting chimeric transcripts supported by both split reads and read pairs (Peer Review Fig. 5c). For example, 40 reads supported a fusion transcript between a L1_Mm transposon and the *Erd1* locus. Another highly scoring example was the *MERV1_LTR-Mtf2*: Genome browser tracks highlighted the chimerism between upstream *MERV1_LTR* sequences and exons within the *Mtf2* locus (Peer Review Fig. 5d). Also, *Mtf2* was almost 1.5 fold upregulated on RNA-seq level. Another striking example was the promoter-associated fusion transcript for *Ube3a*, which itself was found to be around 2-fold upregulated in RNA-seq (Peer Review Fig. 5e).

Comparing our results to previously identified chimeric transcripts in *Setdb1* KO cells (Karimi et al., 2011), none of the five experimentally validated chimeric transcripts expressed in *Setdb1* KO cells could be identified in MPP8 deficient cells. However, it is to be kept in mind that slight differences in depletion kinetics could change the presence of detectable chimeric transcripts. Here, we looked at very early chimerism 6 hours after MPP8 depletion, while removal of SETDB1 requires extended time due to the nature of the gene level knockout system.

While we think that the detected gene-TE chimeric transcripts are highly interesting, our analysis is preliminary and requires further experimental validation and assessment of functional relevance. This analysis would be outside the focus of the current manuscript but will be of interest for a potential follow-up study.

Peer Review Figure 5: Detection of gene-TE chimeric transcripts in MPP8 deficient cells. a) Total number of gene-TE chimeric transcripts in *Mpp8^{MAD};OsTiR1* cells, upon addition of 500 μ M IAA (6 hours) or additional expression of *MPP8^{WT}*. Here, one replicate was considered for the analysis of each cell line. *b)* Total repeat element recurrence in gene-TE chimeric transcripts enriched in *Mpp8^{MAD};OsTiR1* cells treated with IAA for 6 hours. The top 20 enriched repeats are shown. Different repeat families are highlighted in different colors. *c)* Top 10 gene-TE chimeric transcripts enriched in *Mpp8^{MAD};OsTiR1* cells treated with IAA for 6 hours. Thereby, the detection of gene-TE chimeric transcripts was supported by both split reads and read pairs. *d-e)* Genome browser tracks exemplifying the detection of chimeric transcripts between *(d)* *MERV1_LTR* and *Mtf2* and *(e)* *Lx2B2* and *Ube3a*.

4. The authors state on page 13 that “We observed differential regulation of 722 transcripts as early as six hours following MPP8 depletion, among these 396 arising from annotated genes

and the remaining 326 from other annotated elements, such as repeats.” These transcripts and their genomic locations/genes should be included in a supplementary table, divided according to their class.

We have added the requested table containing differentially regulated transcripts and their genomic locations in Supplementary Data 4.

5. Page 14 lines 15-16 The authors state: “Taken together, these results show that DNA methylation and MPP8 independently silence LINE1 expression in metastable mESCs, and that inactivation of both of these repressive mechanisms leads to growth arrest.” Would be nice if the authors commented on the fact that while the DNMT TKO has less DNAm than WT ESCs cultured in 2i/LIF, the DNMT TKO cell growth phenotype in the absence of MPP8 is LESS severe than that observed for ESCs cultured in 2i/LIF in the absence of MPP8 (figure 1).

It is important to notice that the competition-based proliferation assays shown in Figure 1 are not a direct measure of cell speed, as a regular growth curve, but rather an indicator of cell fitness, in particular if cells depleted for a certain gene through sgRNA-mediated targeting are impaired in their fitness compared to the same cell line wildtype for the respective gene. Rather than comparing absolute depletion levels between experiments, cell line intrinsic controls have to be considered as depletion levels of positive control sgRNAs. Relative to *Nat10*, we reach 91% out-competition efficacy (average between all three *Mpp8*-targeting sgRNAs) in MPP8-depleted cells grown in 2i/LIF (Fig. 1e), compared to 70% in *Dnmt* TKO (serum/LIF) cells and 7% in mESCs in serum/LIF (Supplementary Fig. 8d). While we consider the difference between *Dnmt* TKO (serum/LIF) cells and mESCs (serum/LIF) as striking, we hesitate to draw strong conclusions regarding differences between mESCs (2i/LIF) and *Dnmt* TKO (serum/LIF) cells, as the small difference might be a result of e.g. higher transduction efficacies reached in the 2i/LIF experiment (72% GFP+ starting level) compared to 38% in *Dnmt* TKO and hence slightly faster out-competition kinetics (experiments of the same replicate shown in Supplementary Fig. 8d. were conducted simultaneously to make the results as comparable as possible, e.g. to reach similar transduction efficacies, while 2i/LIF experiments were conducted independently).

6. Would the authors care to speculate on the possibility that the N-terminus/chromodomain of MPP8 may be required for binding to SETDB1 due to methylation of ATF7IP, which was previously shown to be methylated (by G9a) and bound by MPP8 via the methylated residue, (Tsusaka et al., Epigenetics & Chromatin, 2018)?

In agreement with the suggested mechanism by Tsusaka et al., we observed impaired recruitment of MPP8 to some of its target loci (Fig. 6b), and have included the following sentence in the text:

“G9a/GLP dKO impacted MPP8 recruitment (Fig. 6b) at some of the investigated MPP8-bound regions, in agreement with the proposed role of G9a in the recruitment of MPP8 via methylation of ATF7IP¹⁵.”

7. The authors state in the Discussion on page 16 that “This observation also raises the exciting possibility that the HUSH complex displays an essential regulatory role in LINE1 control in other hypomethylated cell types, such as germ cells and cancer cells”. *Setdb1* at least has been deleted in both early germ cells (Liu et al, G&D, 2014) as well as growing oocytes (Eymery et al, Development, 2016), where DNAm levels are low, and little upregulation of L1 elements was observed in either. The authors should mention these findings.

We thank the referee for mentioning these studies. We have now discussed the suggested literature in the discussion section of the revised manuscript.

“Interestingly, deletion of *Setdb1* in early germ cells⁵¹ was shown to have little impact on reactivation of LINE1 elements besides low levels of DNA methylation and a clear decrease of H3K9me3 at specific H3K9me3-enriched LINE1 loci. Similarly, no LINE1 de-repression was observed upon removal of *Setdb1* in growing oocytes⁵², suggesting that if HUSH is required for the maintenance of LINE1 repression in these hypomethylated settings, it would be independent of SETDB1.”

8. In Figure 4e, each of the KOed factors should be aligned properly in the horizontal dimension, *Atf7ip* can simply be left blank in the MEF column.

As requested, the KOed factors are now aligned properly and *Atf7ip* was left blank in the MEF column of the revised Fig. 4e.

9. H3K9me3 and DNAm tracks (the later published data in serum/LIF and 2i/LIF) should be added to Figure 4F. Really, it would be nice if these were added to each of the screen shots shown.

We have now added H3K9me3 and published DNAm tracks of mESCs cultured in serum/LIF or 2i/LIF conditions to each screen shot shown.

10. Label other significant dots in the Supplementary Fig 5a.

We have now added the labels for other significant TASOR interaction partners in Supplementary Fig 5a.

11. The culture conditions used for the expts presented in Figures 5 and 6 should be included in the figure/figure legend. This is critical to interpret the data as global DNAm levels differ in mESCs cultured in serum/LIF vs 2i/LIF.

As requested, mESC culture conditions have been added to the figure legends.

12. For genome-browser screen shots with repeatmasker tracks included, where L1 elements are also labelled at the bottom, arrows should be added to indicate which of the repeats shown is actually the labelled/relevant L1 element. Adding genomic orientation of L1 elements would also help to compare the enrichment of H3K9me3 over the 5'-end/promoter or body of L1s in WT vs mutants.

We have now added labels as well as genomic orientation of relevant LINE1 elements in the genome-browser screen shots shown.

Errata:

1. Figure 2e: IAA (500 M) should be corrected to IAA (500 μ M).

2. Figure 4f: “FLAG” is hard to see.

3. Figure 4h: “Refseq genes” should be removed as I believe this panel shows the enrichment of MPP8 around their peak regions.

4. Figure 5c: y-axis is missing.

5. Figure 6a,b: “Suv39h1/h1” should be corrected to “Suv39h1/h2”.

6. Supplementary Fig. 9f is not referred to in the text.

We thank the referee for spotting these errors, they have been corrected in the revised manuscript.

Reviewer #2 (Remarks to the Author):

Muller et al (MS ID#: NCOMMS-20-20955)

In this study, the authors identified a protein named MPP8 as being essential for ground-state pluripotency. MPP8 is a component of the human silencing hub (HUSH) complex and has been suggested to repress LINE1 elements by recruiting the HUSH complex to H3K9me3-rich regions (Tchasovnikarova et al 2015 and Liu et al 2018). Interestingly and unexpectedly, the N-terminus deletion mutant of MPP8 can rescue the growth defect phenotype of the MPP8 depleted ES cells under the ground-stage culture condition (2i+LIF) and also suppress the HUSH-target LINE1 expression, but cannot accumulate on the HUSH-target loci. Furthermore, in the ES cells only expressing the N-terminus deletion mutant of MPP8, transcriptional repression of LINE1 elements is maintained without retaining H3K9me3 levels. Based on those findings, the authors propose that MPP8 silences LINE1 expression and protects the DNA-hypomethylated pluripotent ground state through its association with the HUSH core complex, however, stable chromatin binding and the maintenance of H3K9me3 is not essential for the MPP8-mediated LINE1 repression.

Overall, quality of presented data in this study is high and the manuscript is well written.

The reviewer likes the epitope-tagged approach for biochemical and CHIP studies of endogenous molecules. The findings of the N-terminus and C-terminus deletion mutant of MPP8, MPP8112–858 and MPP81–729 are potentially interesting, for elucidating the molecular mechanism of the HUSH complex-mediated transcriptional silencing. Therefore, the reviewer is supportive for publication of this study in Nat Comm if the authors can address following critical issues.

Major comments,

1) MPP8 (WT and mutants) CHIP-seq and -qPCR analysis for the HUSH-target young LINE1 elements (derepressed copies by MPP8 depletion). This data is absolutely essential to show in this study even if both MPP8112–858 and MPP81–729 lost binding property to the HUSH-target LINE1 elements as shown for other MPP8-target loci such as Kcnq1 and Srm2.

We have now extended Supplementary Fig. 6c and 6d to show CHIP-qPCR analysis on two peaks that overlap LINE1 elements, two non-LINE1 MPP8 binding sites and two negative control loci not bound by MPP8. Moreover, we have performed average plots over the subset of MPP8 peaks, which overlap upregulated LINE1 classes, 6 hours after induction of MPP8 degradation (Peer Review Figure 6). As the results are similar to those for average plots over all MPP8 peaks (Fig. 4h,i), the conclusions drawn in the manuscript remain the same.

Peer Review Figure 6: Enrichment of MPP8 on bulk L1 promoter regions. a, b) Aggregate plot comparing the average MPP8 (h) and FLAG (j) ChIP signal, respectively, over high confidence MPP8 peaks overlapping upregulated LINE1 classes 6 hours after MPP8 removal (n=34) in Mpp8^{mAID}; OsTIR1 cells additionally expressing MPP8^{wt}, MPP8¹⁻⁷²⁹, MPP8¹¹²⁻⁸⁵⁸ after treatment with 500 μ M IAA (16 hours). Input signal (a) and empty parental cells (b), respectively, serve as control.

2) Fig. 4 MPP8 ChIP-seq analysis data and MPP8 WT 55 high confidence peaks.

The authors should describe more about distribution of the MPP8 WT 55 high confidence peaks. Genic, TSS, repeats,... Also, how many of them were mapped to the derepressed LINE1 elements? Supp Fig. 7 may describe this issue, but not enough explanation. Furthermore, to prove the ChIP-seq analysis data, the authors should perform MPP8 ChIP-qPCR analysis of the enriched LINE1 and show it, too.

We have now added the genomic feature distributes of the 55 high-confidence MPP8 binding sites (Supplementary Fig. 5d) as well as a pie chart displaying the percentage of MPP8 peaks overlapping transcriptionally de-repressed LINE1 classes (Supplementary Fig. 7g), and the following passages in the manuscript:

“The majority of MPP8 was found to be located in intergenic regions (60%) followed by introns (22%), and transcription termination sites (11%)”

“Moreover, the majority of high-confidence MPP8 DNA binding sites directly overlapped LINE1 classes upregulated in RNA-seq: 25% of peaks overlapped evolutionary young LINE1 elements with strongest upregulation levels ($FC > 2$, $p_{adj} \leq 0.05$), 24% overlapped older LINE1 classes found to be deregulated ($FC > 1$, $p_{adj} \leq 0.05$) and 13% of peaks overlapped with both youngest and older classes (**Supplementary Fig. 7b**). Together, these observations support a direct role of MPP8 in repressing LINE1 expression.”

3) Fig. 6c and Supp Fig. 9.

3)-1. Related to 1) and 2), H3K9me3 ChIP-qPCR analysis should be done to the derepressed different copies of young LINE1 elements such as L1Md_T and _Gf.

Due to the repeat regions in the LINE1 elements in the genome, especially of the youngest classes as L1Md_T and _Gf, it is unfortunately not technically possible to design ChIP-qPCR primers that target specific L1Md_T or _Gf loci bound by MPP8.

3)-2. Is this specific L1Md_T_dup5018 Lx9_dup 10388 analyzed in Fig. 6c really derepressed L1 in MPP8 depletion? If minor population of entire L1Md_T copies is derepressed by MPP8 depletion, it is difficult to deal with MPP8 targeting or H3K9me3 levels and MPP8-mediated

LINE1 silencing issue because ChIP-seq reads of L1Md_T sequences seem to be multiply mapped regardless of transcriptional status. Should deal with this problem.

As described above, it is not possible to perform locus-specific qPCR analysis on evolutionary young LINE1 elements, however we observed upregulation of these classes on RNA-seq level using agglomerated analysis at already early timepoints after MPP8 degradation and not when using knockout or WT rescue control cell lines. The reviewer is correct in noting that ChIP-seq reads were mapped allowing for multimappers to increase the mappability of these highly repetitive regions. However, as exemplified in the genome browser screen shots, MPP8 and H3K9me3 ChIP-seq peaks extend over regions that are larger than the repeat elements, the average length for MPP8 peaks is indeed 11097 bp, and appropriate controls were used. Hence, this allows us to draw the conclusions presented in the manuscript.

3)-3. Supp Fig. 9E and F (Aggregate plot comparing the average H3K9me3 ChIP-seq signal over all identified MPP8 peaks), these analysis should be done among only the MPP8-targeted LINE1 elements.

As requested, we have added the average profiles over the subset of MPP8 peaks that overlap LINE1 classes significantly upregulated 6 hours after induction of MPP8 degradation (Supplementary Fig. 10). For both 6 hours and 48 hours after MPP8 removal we found similar trends when comparing all MPP8 binding sites, those that overlap upregulated LINE1 classes and those that do not. While we observed upregulation of LINE1 elements already at 6 hours after inducing MPP8 degradation, H3K9me3 levels over MPP8-targeted LINE1 elements were unchanged at this early timepoint. At 48 hours after MPP8 degradation, H3K9me3 levels over MPP8-targeted LINE1 elements were reduced in unrescued cells or cells rescued with either N-terminally or C-terminally truncated mutants. These observations are in agreement with our conclusion that maintenance of H3K9me3 is not required to repress MPP8-regulated LINE1 elements.

3)-4. Currently no clear mechanism how MPP8^{112–858} can suppress LINE1 expression without maintaining H3K9me3 level. At least, the authors should examine other epigenetic marks which have an impact on transcription such as acetylation, H3K4me_{2/3} in comparison with WT and MPP8^{1–729}.

We have now assessed H3K4me₃ and H3K27ac levels using ChIP-qPCR in the MPP8^{mAID};OsTIR1 cells expressing different MPP8 mutants ± IAA (48 hours) (Peer Review Figure 7). In agreement with MPP8 binding sites being mostly located in intergenic and intronic regions and not on promoters or enhancers, H3K4me₃ and H3K27ac levels are low. Moreover, we did not detect any significant changes in H3K4me₃ or H3K27ac levels upon expression of different MPP8 mutants.

Peer Review Figure 7: ChIP-qPCR using an (a) H3K4me3 and (b) H3K27ac antibody, at one promoter region (Dpy30), three enhancer regions (Actin, Nanog, Klf4), two MPP8 binding sites that overlap LINE1 elements (L1_mus3, Kcnq1ot1 locus; L1Md_F2, chr2), two non-LINE1 MPP8 binding sites (Srrm2, Zfp617) and one negative control locus (chr2 desert) in indicated cell lines grown in 2i/LIF.

4) Fig. 6A MPP8 ChIP-qPCR

Same with 1)~3), the MPP8-target LINE1 elements should be examined for the impact of SETDB1 or other H3K9 methyltransferases depletion. Also, “Data show one representative experiment (n = 1)” is not sufficient to make their conclusion. Provide additional evidences.

We have now extended Fig. 6a and 6b showing two peaks that overlap LINE1 elements, two non-LINE1 MPP8 binding sites and two negative control loci not bound by MPP8. The panel now represents two biological replicates (n = 2).

5) How about LINE1 depression in the MPP8 depleted ES cells under the culture condition with single addition of GSK inhibitor which did not induce growth defect by MPP8 depletion? This may provide additional insight about the role of MPP8 for ground-state pluripotency and LINE1 regulation.

To gain additional insight into LINE1 de-repression in MPP8-mediated maintenance of ground state pluripotency, we have now investigated L1orf2 expression levels upon MPP8 depletion in different culture conditions (Supplementary Fig. 8b). We found a correlation between the level of LINE1 de-repression and impaired stem cell self-renewal capacity in MPP8-depleted cells: addition of the GSK inhibitor showed the lowest levels of L1orf2 de-repression, consistent with the only minor observed growth defect (Supplementary Fig. 3d). Both the expression level of L1orf2 and the proliferation capacity were indeed similar to those seen for serum/LIF only cultured mESCs. Addition of either the MEK inhibitor alone or both of the inhibitors, in contrast, led to stronger L1orf2 expression in MPP8-depleted cells (Supplementary Fig. 8b) concomitant with greater proliferation defects (Supplementary Fig. 3d).

We have added the following paragraph to the text:

“Moreover, while addition of the GSK inhibitor to serum/LIF culture did not increase L1orf2 expression in MPP8-depleted cells further, serum/LIF culture supplemented with MEKi or 2i, for which we observed lower proliferation capacity (Supplementary Fig. 3d), yielded increased L1orf2 expression levels following MPP8 removal (Supplementary Fig. 8b). These results are in line with the hypothesis that mESC self-renewal is impaired when LINE1 expression exceeds a certain level.”

6) MPP81-729

RNA-seq analysis at 48 hours post auxin treatment, the phenotype of MPP8 depleted cells and MPP8 depleted cells complemented with MPP81-729 is quite different. Later cells induced 10x more up- or down-regulated transcripts (Non rescued: 314, MPP81-729: 4481). Furthermore, MPP81-729 showed more strong impact than simple MPP8 depletion, or MPP81-522 and MPP81-188 rescued cells (Fig. 2c and Fig. 3d), suggesting that MPP81-729 is not a simple recessive mutant, but a dominant negative one. **The authors should discuss this in the text.**

We thank the referee for the comment. We now discuss the possibility of a potential dominant negative effect in the revised manuscript.

We have added the following sentence to the text:

“Together with the severe proliferative defect of MPP8¹⁻⁷²⁹-expressing cells (**Fig. 3d**) and the observed destabilization of the HUSH core complex in presence of MPP8¹⁻⁷²⁹ (**Supplementary Fig. 5b**), the stronger transcriptional impact compared to deletion of MPP8 in wildtype cells might additionally suggest a dominant-negative role of C-terminally truncated MPP8.”

Minor comments,

1) In introduction, “Moreover, MPP8 has been shown to interact with multiple epigenetic silencing proteins, including the H3K9 mono- and di-methyltransferase proteins GLP/G9a^{11,12}, DNA methyltransferase DNMT3A^{11,12} and histone deacetylase SIRT1¹³.”

ATF7IP should be included. ATF7IP methylated by G9a/GLP is also shown to bind MPP8 (Tsusaka et al 2018).

We have now added ATF7IP to the list of known MPP8 interaction partners:

“Moreover, MPP8 has been shown to interact with multiple epigenetic silencing proteins, including the H3K9 mono- and di-methyltransferase proteins GLP/G9a^{12,13}, DNA methyltransferase DNMT3A^{12,13}, histone deacetylase SIRT1¹⁴ and ATF7IP, a known binding partner of H3K9 tri-methyltransferase SETDB1¹⁵.”

REVIEWER COMMENTS

Reviewer #1 (Remarks to the Author):

The revised manuscript is much improved and the authors have addressed each of my concerns sufficiently. There are a number of typos, including, incorrect references to figure panels, that should be corrected before publication.

Also, another relevant HUSH paper, previously on bioRxiv, now published in Nature Communications (<https://www.nature.com/articles/s41467-020-18761-6>), should be cited here and contrasted with the results of this study in the discussion, as the issue of MPP8 (And Tasor) domains necessary for transgene repression are addressed in Douse et al. as well. Fortunately, the results seem consistent between the two studies.

Minor issues to address:

Line 292:

In agreement with being partners of MPP8 in the HUSH complex, both proteins were found to associate with MPP8-bound regions (Fig. 4f and Supplementary Fig. 5f). "5f" should be "5g".

Line 368: "Moreover, the majority of high-confidence MPP8 DNA binding sites directly overlapped LINE1 classes as shown upregulated by RNA-seq", should be "classes shown as upregulated..."

Line 371;

"overlapped older LINE1 classes found to be deregulated ($FC > 1$, $p_{adj} \leq 0.05$) and 13% of peaks overlapped with both youngest and older classes (Supplementary Fig. 7b)." SHOULD BE Supplementary Fig. 7g.

Line 380 "RNA-seq analysis at 48 hours post auxin treatment showed differential regulation of 314 transcripts (Fig. 5h), 290 of which were re-repressed by wildtype MPP8, including four LINE1 classes (Fig. 5i, Supplementary Fig. 7f). THE AUTHORS NEED TO CHECK THE FIGURE PANEL REFERENCES CAREFULLY. THEY ARE NOT CORRECT.

Reviewer #2 (Remarks to the Author):

The reviewer satisfied with majority of authors' responses. The revised manuscript improved a lot. However, the mechanistic issue of MPP8(112–858)-mediated L1 silencing remains to be addressed appropriately. The reviewer requests that the authors should deal with this issue again.

The reviewer's original comment: 3)-4. Currently no clear mechanism how MPP8112–858 can suppress LINE1 expression without maintaining H3K9me3 level. At least, the authors should examine other epigenetic marks which have an impact on transcription such as acetylation, H3K4me2/3 in comparison with WT and MPP81–729.

Authors' response: We have now assessed H3K4me3 and H3K27ac levels using ChIP-qPCR in the MPP8mAID;OstTIR1 cells expressing different MPP8 mutants \pm IAA (48 hours) (Peer Review Figure 7). In agreement with MPP8 binding sites being mostly located in intergenic and intronic regions and not on promoters or enhancers, H3K4me3 and H3K27ac levels are low. Moreover, we did not detect any significant changes in H3K4me3 or H3K27ac levels upon expression of different MPP8 mutants.

The reviewer's further comment: Peer Review Fig. 7 shows no increase of active chromatin modifications on the derepressed L1s (L1_mus3 in Kcnq1ot1 locus and L1md_F2 on chr2) were observed in Mpp8 depleted cells, which implicates that location of the primer sets used in this analysis are not suitable for investigating how MPP8(112–858) represses L1 expression. The reviewer understands that it is difficult to design primers to analyze chromatin modification of specific L1 copy due to highly similar sequences among the L1 elements. However, it is quite important to know which copies of L1 are repressed by MPP8 wt and MPP8(112–858) for unraveling the function of N- and C-terminus of MPP8. The authors stated that MPP8 binding sites are mostly located in intergenic and intronic regions and not on promoters or enhancers. If so, how MPP8 binding to target genes and L1 elements seen for full-length MPP8 plays a role for their silencing? The authors already described dynamics of H3K9me3 on the L1-mus3 in Kcnq1ot1 locus by using H3K9me3 ChIP-seq data (Fig. 6c). Even if the MPP8 binding site to this element is out of the promoter or enhancer region, it seems that the L1-mus3 is entirely covered with H3K9me3 mark (spreading from the binding site?). Thus, it is also possible that active epigenetic marks on the promoter or enhancer of this element can be affected by the MPP8 binding (targeting). So, if the authors apply same ChIP-seq analysis, they can also address how active epigenetic marks are modulated by MPP8 depletion and complemented with wt, N- or C- terminus deletion mutant of MPP8 on entire MPP8 targeted L1 copies (derepressed by MPP8 depletion) which is based on the authors' logics. The authors should do this. ATAC-seq analysis is also welcome. Alternatively, if it is possible to identify derepressed L1 copies from RNA-seq data in MPP8 depleted cells, comparison analysis of MPP8 enrichment, H3K9me3 and active chromatin modifications in each cells expressing either wt or N- or C- terminus deletion mutant of MPP8 around those L1 copies is also acceptable.

Point-to-point response to the two Referees

REVIEWER COMMENTS

Reviewer #1 (Remarks to the Author):

The revised manuscript is much improved and the authors have addressed each of my concerns sufficiently. There are a number of typos, including, incorrect references to figure panels, that should be corrected before publication.

We thank the reviewer for the positive evaluation of our revised manuscript. All suggested edits have been incorporated.

Also, another relevant HUSH paper, previously on bioRxiv, now published in Nature Communications (<https://www.nature.com/articles/s41467-020-18761-6>), should be cited here and contrasted with the results of this study in the discussion, as the issue of MPP8 (And Tasor) domains necessary for transgene repression are addressed in Douse et al. as well. Fortunately, the results seem consistent between the two studies.

We thank the reviewer for this comment. We have now compared our findings related to the requirement of MPP8 domains on endogenous LINE1 repression and mESC self-renewal to the results presented by Douse et al. in the discussion section:

“When our manuscript was under review, another study was published addressing the functional requirement of MPP8 domains in HUSH-mediated repression of transgenes. Consistent with our structure-function analysis addressing the requirement of MPP8’s domains for endogenous LINE1 element repression and stem cell self-renewal, genetic complementation of *Mphosph8* knockout cells with the human MPP8^{500–860} protein was found to silence a GFP reporter construct, whereas MPP8^{1–728} did not. Hence, the requirement of MPP8’s C-terminal region is conserved between both its silencing activity on endogenous LINE1 elements and that on transgenic insertions, while the chromodomain is dispensable for both functions.”

Minor issues to address:

Line 292:

In agreement with being partners of MPP8 in the HUSH complex, both proteins were found to associate with MPP8-bound regions (Fig. 4f and Supplementary Fig. 5f). “5f” should be “5g”.

Line 368: “Moreover, the majority of high-confidence MPP8 DNA binding sites directly overlapped LINE1 classes as shown upregulated by RNA-seq”, should be “classes shown as upregulated...”

Line 371;

“overlapped older LINE1 classes found to be deregulated (FC > 1, padj <= 0.05) and 13% of peaks overlapped with both youngest and older classes (Supplementary Fig. 7b).” SHOULD BE Supplementary Fig. 7g.

Line 380 “RNA-seq analysis at 48 hours post auxin treatment showed differential regulation of 314 transcripts (Fig. 5h), 290 of which were re-repressed by wildtype MPP8, including four

LINE1 classes (Fig. 5i, Supplementary Fig. 7f). THE AUTHORS NEED TO CHECK THE FIGURE PANEL REFERENCES CAREFULLY. THEY ARE NOT CORRECT.

Thank you for spotting these errors, they have been corrected in the revised version of our manuscript.

Reviewer #2 (Remarks to the Author):

The reviewer satisfied with majority of authors' responses. The revised manuscript improved a lot. However, the mechanistic issue of MPP8(112–858)-mediated L1 silencing remains to be addressed appropriately. The reviewer requests that the authors should deal with this issue again.

We thank the reviewer for the overall positive evaluation of our revised manuscript. We have addressed the remaining question focusing on chromatin-based modulation of LINE1 transcription in MPP8 wildtype and mutant cells in the below paragraph and added Supplementary Figure 8 to the manuscript.

The reviewer's original comment: 3)-4. Currently no clear mechanism how MPP8112–858 can suppress LINE1 expression without maintaining H3K9me3 level. At least, the authors should examine other epigenetic marks which have an impact on transcription such as acetylation, H3K4me2/3 in comparison with WT and MPP81–729.

Authors' response: We have now assessed H3K4me3 and H3K27ac levels using CHIP-qPCR in the MPP8mAID;OsTIR1 cells expressing different MPP8 mutants \pm IAA (48 hours) (Peer Review Figure 7). In agreement with MPP8 binding sites being mostly located in intergenic and intronic regions and not on promoters or enhancers, H3K4me3 and H3K27ac levels are low. Moreover, we did not detect any significant changes in H3K4me3 or H3K27ac levels upon expression of different MPP8 mutants.

The reviewer's further comment:

Peer Review Fig. 7 shows no increase of active chromatin modifications on the derepressed L1s (L1_mus3 in Kcnq1ot1 locus and L1md_F2 on chr2) were observed in Mpp8 depleted cells, which implicates that location of the primer sets used in this analysis are not suitable for investigating how MPP8(112–858) represses L1 expression. The reviewer understands that it is difficult to design primers to analyze chromatin modification of specific L1 copy due to highly similar sequences among the L1 elements.

However, it is quite important to know which copies of L1 are repressed by MPP8 wt and MPP8(112–858) for unraveling the function of N- and C- terminus of MPP8. The authors stated that MPP8 binding sites are mostly located in intergenic and intronic regions and not on promoters or enhancers. If so, how MPP8 binding to target genes and L1 elements seen for full-length MPP8 plays a role for their silencing? The authors already described dynamics of H3K9me3 on the L1-mus3 in Kcnq1ot1 locus by using H3K9me3 CHIP-seq data (Fig. 6c).

Even if the MPP8 binding site to this element is out of the promoter or enhancer region, it seems that the L1-mus3 is entirely covered with H3K9me3 mark (spreading from the binding site?). Thus, it is also possible that active epigenetic marks on the promoter or enhancer of this element can be affected by the MPP8 binding (targeting). So, if the authors apply same CHIP-seq analysis,

they can also address how active epigenetic marks are modulated by MPP8 depletion and complemented with wt, N- or C- terminus deletion mutant of MPP8 on entire MPP8 targeted L1 copies (derepressed by MPP8 depletion) which is based on the authors' logics. The authors should do this. ATAC-seq analysis is also welcome. Alternatively, if it is possible to identify derepressed L1 copies from RNA-seq data in MPP8 depleted cells, comparison analysis of MPP8 enrichment, H3K9me3 and active chromatin modifications in each cells expressing either wt or N- or C- terminus deletion mutant of MPP8 around those L1 copies is also acceptable.

As requested, we have now performed ChIP-seq analysis of H3K4me3 and H3K27ac to address how these permissive epigenetic marks are modulated on a genome-wide level following MPP8 depletion or in response to complementation with N- or C-terminal MPP8 mutants (**Supplementary Fig. 8**).

For this purpose, we assessed the enrichment of H3K4me3 at the transcription start site (± 2 kb) of all annotated LINE elements and further selected LINE1 elements that belong to classes which showed significant upregulation ($FC > 1.5$, $p < 0.05$) in the RNA-seq data as well as an increase of at least two-fold in H3K4me3 at 48 hours after MPP8 degradation ($n = 512$).

The results from these experiments showed that H3K4me3 levels at LINE1 TSSs that were found to increase upon MPP8 depletion were efficiently rescued upon reintroduction of MPP8 wildtype as well as the N-terminal deletion mutant, while expression of the C-terminal deletion mutant showed no rescue (**Supplementary Fig. 8a**). We found that H3K27ac levels reflected the changes observed for H3K4me3, although less pronounced (**Supplementary Fig. 8b**).

To further understand which LINE1 copies are repressed by MPP8^{wt} and MPP8¹¹²⁻⁸⁵⁸, we looked more closely at LINE1 length distribution (**Supplementary Fig. 8c**) and contribution of specific LINE1 classes (**Supplementary Fig. 8d**). We found that MPP8 was significantly more likely to repress longer LINE1 elements (**Supplementary Fig. 8c**). In contrast, the proportion of evolutionary young classes (L1Md_T, L1Md_A) was similar between MPP8-repressed LINE1 elements and all LINE1 elements part of activated classes (**Supplementary Fig. 8d**). These data suggest that longer, and hence more likely to be transcribed LINE1 classes, acquire a more permissive chromatin landscape at their TSSs in response to MPP8 degradation or its functional inactivation, i.e. C-terminal truncation.

In conclusion, the observed epigenomic changes corroborate the RNA-seq data, for which we saw efficient LINE repression upon expression of MPP8^{wt} and MPP8¹¹²⁻⁸⁵⁸, but not MPP8¹⁻⁷²⁹.

We have included the following paragraph in the manuscript:

“By profiling permissive chromatin post translational modifications in the different cell lines using ChIP-seq, we found that H3K4me3 levels at LINE1 TSSs that were increased upon MPP8 depletion were efficiently rescued upon reintroduction of MPP8 wildtype as well as the N-terminal deletion mutant, while expression of the C-terminal deletion mutant did not rescue (**Supplementary Fig. 8a**). A similar change, although less strikingly, was also observed for H3K27ac levels (**Supplementary Fig. 8b**). We also found that LINE1 elements modulated by the loss of MPP8 were significantly longer compared to all LINE1 elements belonging to these classes (**Supplementary Fig. 8c**), while the contribution of young L1Md_T/A elements to both activated and all elements in these classes was similar (**Supplementary Fig. 8d**). Since evolutionary young, full-length LINE1 elements, which maintain intact promoter regions, are more likely to be transcribed compared to shorter elements that have often been rendered inactive through truncation³⁴, regulation by MPP8 might be particularly required to safeguard the pluripotent

epigenome at these transcription-permissive full-length LINE1 elements. Consistent with this hypothesis, we found young LINE1 elements to be de-repressed by MPP8 loss (**Fig. 5h, Supplementary Fig. 7f**). We conclude that MPP8 depletion or its C-terminal truncation in ground-state mESCs result in increased levels of permissive chromatin modifications at transcription start sites of transcription-permissive LINE1 elements and their enhanced expression.”

REVIEWERS' COMMENTS

Reviewer #2 (Remarks to the Author):

The authors properly responded to the reviewer's further comment. No more comment.

Point-to-point response to the Referees

We thank the referees for recommending the manuscript for publication.